# Heavy metal footprints in landfill-proximate soils of Jashore, Bangladesh: An index-based risk assessment

**Sanjida Sultana Santa[1], Md Kamal Hossain**[2,3]*, **Kowshik Das Karmaker**[4], **Mohammad Moniruzzaman**[2,3], **Md. Harunor Rashid Khan**[1]

**1** Department of Soil, Water and Environment, University of Dhaka, Dhaka, Bangladesh, **2** BCSIR Laboratories Dhaka, Bangladesh Council of Scientific and Industrial Research (BCSIR), Dhaka, Bangladesh, **3** Central Analytical Research Facilities (CARF), Bangladesh Council of Scientific and Industrial Research (BCSIR), Dr. Qudrat-E-Khuda Road, Dhanmondi, Dhaka, Bangladesh, **4** Department of Oceanography, University of Dhaka, Dhaka, Bangladesh

* kamalbcsir@gmail.com; kamalhossain@bcsir.gov.bd

## Abstract

Disposal of the household originated and industrial hazardous materials with municipal solid waste (MSW) into the open dumping sites is the usual practice in Bangladesh which raises concerns regarding the pollution of the regions adjacent to these landfill sites. The study evaluated the impact of landfill techniques and environmental factors on soil contamination by potentially toxic elements and the corresponding environmental hazards. This study assessed the potential ecological hazards linked to a landfill site in southwestern Bangladesh. ICPMS and AAS were used to detect eleven heavy metals in soils from the landfill. This analysis aimed to uncover the origin, degree of contamination, geographical representation, and the environmental and human health hazards connected with these metals. Mean levels of the metals in mg/kg were As ($12.02 \pm 4.40$); Hg ($0.611 \pm 0.441$); Cd ($0.606 \pm 0.487$); Pb ($37.5 \pm 14.08$); Cr ($46.9 \pm 10.06$); Zn ($260.7 \pm 201.0$); Co ($16.03 \pm 3.02$); Ni ($39.5 \pm 11.6$); Cu ($260.7 \pm 55.79$); Mn ($613.2 \pm 189.6$); Fe ($26087 \pm 4396$). This investigation discovered significant amounts of As, Hg, Cd, Pb, Zn, Cu, and Mn relative to established standards. The ecological risk assessment index revealed that Cd exhibited the most significant level of danger, whereas Cu and As demonstrated a moderate level of danger. The health index values indicate that the levels of As and Zn in adults and children exceeded the acceptable limit, while the levels of Pb, Cr, Fe, Cu, Ni, and Co among children came close to the borderline. The TCR values of Ni for children are beyond the permissible level. PCA revealed that the majority of heavy metals are derived from human activity. Consequently, administration strategies for remediating these open dumps nationwide should include continuous monitoring and surveillance of neutralized dumpsites to avert further contamination.

**Data availability statement:** All relevant data are within the paper and its Supporting Information files.

**Funding:** This study was partially funded by the Research and Development (R &D) project of Bangladesh Council of Scientific and Industrial Research (BCSIR) and also partially funded by the special research grant no-39.00.0000.006.99.026.24.65 (SRG-243321; 2024-25) of the Ministry of Science and Technology, the Peoples' Republic of Bangladesh, there was no additional external funding received for this study. The funders had no role in study design, data collection and analysis, decision to publish, or preparation of the manuscript.

**Competing interests:** There is no Competing Financial Interest.

## Introduction

Developing nations typically generate waste at a high rate. The exponential rise in urbanization, growing economy, population, and improved lifestyles are the primary factors contributing to the swift increase in waste production, leading to many socio-economic and ecological challenges [1]. Globally, there is an increasing trend in the rate at which waste is produced. Owing to rapid population growth and urbanization, the amount of waste produced each year is projected to escalate by 73% from 2020 levels, reaching 3.88 billion tonnes by 2050 [2]. Urban regions of Bangladesh generate around 25,000 tons of solid garbage daily, equating to 170 kilograms (kg) per inhabitant annually [3]. The waste volume was 6,500 tons in 1991 and 13,300 tons in 2005, effectively doubling over a 15-year period [4]. Rapid urbanization, alterations in the living conditions of urban residents, and the nation's economic shift towards a middle-income status will result in a continuous growth in waste volume. The per capita daily urban solid waste generation is anticipated to rise to 0.60 kg by 2025, up from 0.49 kg in 1995 [3].

Waste is disposed of without any physical segregation of hazardous materials [5]. Landfill waste disposal is the most favored method compared to alternatives like recycling, incineration, and composting, due to its low cost, ease of operation, and the simplicity of the technologies employed in solid waste management [6]. In open-air dumps, waste is deposited on unprotected soil without gas and leachate collection systems, thereby endangering the surrounding environment due to air, water, and soil pollution resulting from the absence of structures designed to mitigate harmful emissions [7]. The leachate produced in these insufficient deposits is the primary vector of pollutants arising from waste decomposition and rainwater percolation through the landfill; it possesses a complex composition, characterized by a high organic load and various pollutants, which can lead to environmental contamination when it permeates unprotected soil, as observed in dumpsites [8]. In addition, waste burning generates several hazardous chemicals, such as trace, significant, or heavy metals, which are perilous due to their persistence and toxicity [9]. Metals are disseminated by the dumping of bottom ash and the dispersion of fly ash around landfills, ultimately contaminating the surrounding soil [10,11]. Landfills are typically located adjacent to agricultural grounds in the rural and suburban regions of Bangladesh. Heavy metals are notable among leachate pollutants due to their contaminating and bioaccumulative properties. In addition to being a threat to water and soil, they can jeopardize food safety, impact regional crops, and pose a health concern, particularly for individuals residing near the waste disposal site [12,13]. Consequently, the quantities of heavy metals in the soil adjacent to the landfill must be perpetually monitored, even post-inactivation, as leachate is produced and influences the environment over an extended duration [14].

Metals are progressively collected by foodstuffs cultivated near the landfills, ultimately presenting significant risks to human health [15]. Additionally, heavy metals influence the flora consumed by humans and animals. Plants cultivated in soil contaminated with heavy metals have exhibited significantly impaired growth, performance, and yield [16], although some trace elements are necessary for plant

development. Heavy metals adversely impact soil health by inhibiting normal microbial activity and the breakdown of organic pollutants [17]. Ultimately, soil pollution by heavy metals produces multifaceted effects that influence soil microbes and human health.

Numerous studies have investigated heavy metal concentrations and the impacts of metal pollution in areas surrounding chemical industries, tanneries, and other industrial activities, with the majority conducted in and around Dhaka city. Currently, studies on metal contamination in the soil surrounding landfills in Bangladesh are limited. Karim et al. [18] first documented metal pollution in the soil surrounding landfills regions in the Dhaka and Khulna districts of Bangladesh. Their findings revealed a somewhat elevated level of metal compared to the global average values in the study area. Saha et al. [19] studied the concentration patterns of 5 metals in dumping and non-dumping sites during the rainy season randomly in Khulna city and reported elevated metal concentrations at dumping sites. Alongside the monitoring of soil concentrations, pollution indicators and risk assessments have been utilized to safeguard environmental quality and public health in the vicinity of landfills.

Previous studies have investigated only a limited number of metals, specifically some trace and heavy metals, in the soils surrounding landfills. None of the studies assessed a total of 11 metal concentrations in four directions around the landfills. It is noteworthy that the origin of metals was not evaluated in prior investigations. Identifying pollution sources is a crucial step in pollution mitigation, and multivariate statistics, including correlation analysis and factor analysis, are extensively employed for this objective [20]. To distinguish between natural and anthropogenic sources of metal contamination and to quantify the extent of such contamination, various indices, including the contamination factor, enrichment factor, geo-accumulation index, and pollution load index, have demonstrated their utility [21]. In this sense, the study aimed to (i) quantify the concentrations of 11 metals in composite soil samples from landfills surroundings in the four directions; (ii) evaluate metal pollution through the enrichment factor, geo-accumulation index, contamination factor, and pollution load index; and (iii) elucidate the interrelationships among metals, and sources of metals via correlation and factor analysis; iv) evaluate the human health hazards linked to different functional areas utilizing the dose-response model. Consequently, it is anticipated that the findings of this study will furnish data to aid in the formulation of strategies and policies concerning the management of landfills, thereby mitigating harm to the environment and public health.

## Materials and methods

### Study area

The Jashore Pourosova landfill site is situated in the Southwestern area of Bangladesh (at coordinates 89°14′29.33″E and 23°09′45.33″N) (Fig 1) on the bank of the Bhairab River. This location falls within the subtropical monsoon climate. The region experiences heavy precipitation throughout the rainy season, with an average yearly rainfall of approximately 1537 mm. In Jashore, the rainy season is characterized by high temperatures, high humidity, and overcast skies, while the summer season is often characterized by mild temperatures and clear skies. The average minimum and highest temperatures during both winter and summer are approximately 15.3°C and 38.3°C, respectively. The surface geology of the research region is primarily composed of deltaic deposits that were formed during the Holocene period. The delta sediments refer to the deposition of sedimentary material occurring on the dynamic delta region located to the south of the Ganges River. Most of the research area is composed of deltaic silt [22]. The district has a population of around 2.075 million, with over half of its residents involved in agricultural operations and allied occupations. Bhairab and Kopotakkho Rivers are the primary rivers passing through this area [23].

The prevailing wind direction is southwest. The region is distinguished by the presence of silty clay and clay loam soils. The location is encompassed by densely populated regions, comprising primarily educational institutions and residential areas. Additionally, there are other industrial factories located near the landfill site. The landfill site commenced operations in 2018 with a daily capacity of 30 tonnes. However, the amount of incoming garbage currently exceeds this capacity, reaching 50 tonnes per day.

                                                                    

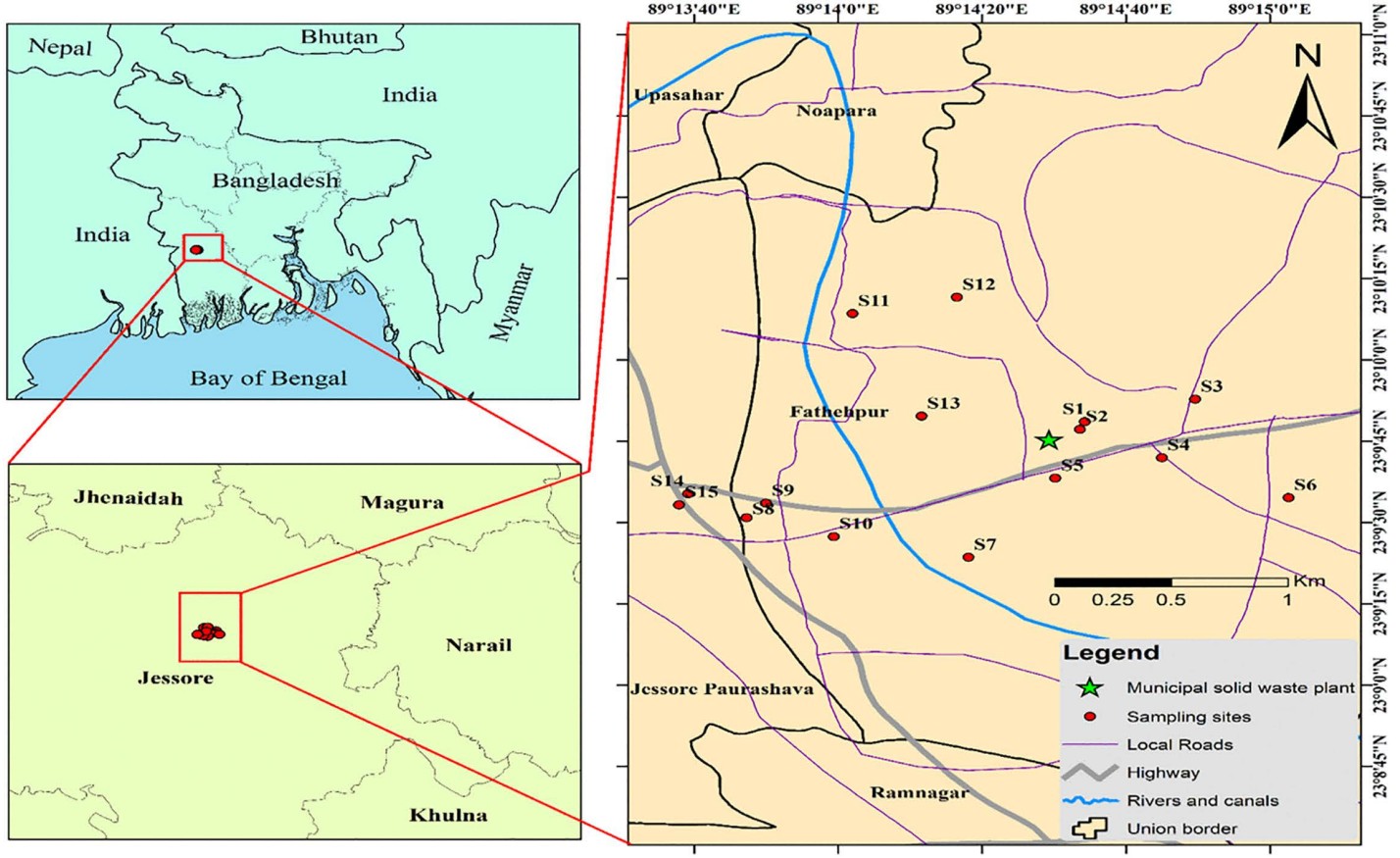

**Fig 1. Study area showing the locations around the landfill site from where soil samples were collected (Source: Map generated by the authors using ArcMap 10.8 software).**

## Soil sampling

In October 2023, a total of 15 surface composite soil specimens (0–20 cm) were obtained surrounding the landfill in four directions. Each composite sample was prepared from 3 sub-samples (triplicate) collected at each site and the sub-samples were thoroughly mixed to obtain one homogeneous composite sample, which reduces localized heterogeneity and analytical uncertainty. The sampling design followed a stratified approach based on land-use types and directional gradients around the landfill. The specimens close to the landfill site were located below the site and above the Bhairab River. Agricultural land was situated behind the landfill site, and near a school and residential area. The samples were collected from the agricultural land, residential area, wild land, and school field (S1 Table). All samples were collected from publicly accessible areas and with the consent of the local landowner. Farmers voluntarily granted access to their land after being informed about the research objectives. No formal permits were required as the study does not involve any protected area or regulated activities. Every sample procedure was conducted with integrity, causing the least disturbance to the environment and human health.

Each specimen (>1 kg in weight) was collected using a metallic steel manual shovel and specimens were taken in triplicate from the sample locations. The soil specimens were enclosed in zip lock bags and labeled with the corresponding date and location identification to prevent contamination and deterioration and then transported to the laboratory for assessment. GPS was utilized to designate the sampling locations. The samples were exposed to natural air drying for

4–6 days. Thereafter, all significant impurities such as roots, plastics, wires, and bricks, wooden pieces were manually discarded. Following a complete homogenization and crushing process, using a pestle and clean mortar, the specimens were then sieved using brass sieves with a mesh diameter of 2 mm. After that, the specimens were sealed in airtight zip-lock plastic containers and stored at 4°C for evaluation [23]. Subsequently, 1 gram of soil samples was taken from each of the 15 stations. In all samples, 10 mL of a solution containing 65% $HNO_3$ was added. However, for the blank, only 10 mL of $HNO_3$ was used [24]. The specimens were placed under a watch glass and kept undisturbed the entire night for pre-digestion. After approximately 12 hours, all specimens were placed on a hot plate with a starting temperature of 80°C [25]. The watch glasses were taken off after the temperature reached 180°C. All specimens were taken off the hotplate once all added acids evaporated. Upon cooling down, 5 mL of $HClO_4$ was added to all of them. The mixture was then reheated on the hotplate until the beaker had just two to three milliliters of solution. Next, the samples were cooled to ambient temperature and then moved to a 50 mL volumetric flask by rinsing them with deionized water [26]. Afterward, the samples underwent filtration using Whatman filter paper 41 which had a diameter of 125 mm, and were then preserved in opaque plastic bottles. The specimens were subsequently stored in the laboratory at a temperature of 4 ° C till elemental analysis was conducted.

From sample collection to laboratory evaluation, we made every effort to preserve the high caliber of the analysis. Sample preparation was carried out with analytical-grade reagents including $HNO_3$, $HClO_4$, $H_2O_2$, and HCl (Sigma-Aldrich, Merck, Switzerland) and deionized water (electricity conductance 0.1 μScm$^{-1}$, resistivity 17.9 million ohm-cm at 25°C). Before being utilized, the measuring flask, pipette, and other equipment in the research procedure were cleaned and standardized by immersing in 10 percent by volume $HNO_3$ for a full night and then rinsing in deionized water [27]. The computerized electric scale used to weigh the samples was the GR-200 from Japan.

## Sample analysis

The quantification of heavy metals was conducted using the Shimadzu AA-7000 model F-AAS followed by inductively coupled plasma-mass spectrometry (ICP-MS) with a PerkinElmer NexION 2000 model equipment from the United States. The nebulized sample is introduced into the central region of the argon field of the ICP-MS system at an injection rate of approximately 0.35 ml/min. A quadrupole mass detector was used to measure the mass-to-charge ratio of metal ions produced in a high-temperature plasma. A standard calibration curve at parts per billion (ppb) levels was established with ICP-multi-element reference solution XIII from Merck, Germany.

## Quality control and assurance

The glassware utilized in the present study underwent thorough disinfecting and rinsing with HCl and water to lessen the risk of pollution throughout specimen preparation and experimental processes. Before calibration with standards, the functioning of the ICPMS instrument had been verified using the NexION Setup solution from PerkinElmer, USA. A calibration deviation of five points was attained with an $R^2$ value greater than 0.9995, covering a fluctuating range of 1.0−25 ng mL$^{-1}$. Instrument stability was confirmed through a minimum of three duplicates. We evaluated the upper limit of the relative standard deviation (RSD) to be between 5% and 8%. The limit of detection (LOD) and limit of quantification (LOQ) for each metal were determined as three and ten times the signal-to-noise ratio, respectively. The detailed LOD and LOQ values for all analyzed metals are provided in S2A Table in S2 Table.

The recovery percentage of significant metals in the specimen was evaluated by utilizing a standard reference material (SRM-1566b) provided by the National Institute of Standards and Technology (NIST). The results indicated a recovery range of 96–103.7% for these elements (S2BTable in S2 Table). As an element of the quality assurance process, the approach involved doing blanks, SRM recoveries, and spike recoveries to reduce inaccuracies. The quantities of metals were adjusted based on the results of recovery and blank sample analysis.

## Statistical analysis

The statistical evaluation and graph formatting for this study were conducted using Microsoft Excel 2019, Origin 2024, ArcMap 10.8, and IBM-SPSS Statistics 26. Descriptive statistical parameters, including minimum (Min), maximum (Max), mean, standard deviation (SD), coefficient of variation (CV), skewness, and kurtosis were calculated to summarize the distribution and variability of heavy metal concentrations. To understand the statistical characteristics of the specimens, box plots were created using Origin 2024 to visualize data dispersion and identify potential outliers. To assess whether the mean heavy metal concentrations significantly differed from shale background values, one-sample Student's t-tests were performed using IBM SPSS Statistics 26. Multivariate analyses, specifically principal component analysis (PCA), cluster analysis, and correlation matrix were conducted using IBM SPSS Statistics 26 to examine relationships among metals and identify potential sources of contamination. The study area map and geological maps of various metals across the stations were created using ArcMap 10.8. Spatial maps of heavy metals and risk indices were generated using Inverse Distance Weighted (IDW) interpolation method. All comparisons presented are descriptive and based on observed concentration patterns and reference values.

## Index for assessing soil contamination

**Contamination factor (CF).** The Contamination Factor (CF) assesses the metal's capacity to contribute to the contamination level caused by a single component. This is said as follows:

$$CF = \frac{C_m}{C_{ref}} \qquad (1)$$

Here, $C_m$ = the amount of the metals under study in the soil, and $C_{ref}$ is the crustal abundance levels from [28] were used as shale values in this context. Shale values were used as background reference concentrations due to the absence of a suitable local control site with comparable soil properties but unaffected by anthropogenic activities.

The soil contaminating levels in the present investigation were categorized into four groups according to the CF [29,30] (S3 Table).

**Pollution Load Index.** Pollution Load Index (PLI) can evaluate the combined effect of different levels of heavy metal contamination at each sampling site.

$$PLI = \sqrt{CF_1 \times CF_2 \times CF_3 \times \ldots \ldots \times CF_n} \qquad (2)$$

The CF represents the contamination factor for every metal, whereas n is the entire amount of metals evaluated in every specimen. A PLI value of more than one indicates the presence of pollution, whereas a value less than one shows the absence of pollution load [31].

**Enrichment factor.** Enrichment Factor (EF) was commonly used to determine the human-made origin of metallic components. Fe was frequently used as a benchmark for estimating the enrichment factor of soil heavy metal contaminants because of its negligible or absent influence on soils [32]. In this investigation, Fe served as the reference element for this inquiry due to its relatively significant concentration on the Earth's surface.

The EF computation is represented by the equation below. The equation is:

$$EF = \frac{\left(\frac{C_s}{C_r}\right)_{sample}}{\left(\frac{C_s}{C_r}\right)_{background}} \qquad (3)$$

The amount of each of the components of interest is denoted as $C_s$, while $C_r$ represents the amount present of the reference element utilized for standardization. An element with an EF value ranging from 0.05 to 1.5 is most likely to be of

natural or crustal origin [33]. The contamination level of the components being examined can be classified into four groups [34] depending on the EF, as displayed in the table (S3 Table).

**The Geo-accumulation Index ($I_{geo}$).** The Geo-accumulation Index ($I_{geo}$) quantifies the level of metal contamination in soil. The $I_{geo}$ was computed utilizing the equation devised by [35] and subsequently employed by other investigators [36]. The formula:

$$I_{geo} = log_2 \left( \frac{C_m}{1.5 \, B_n} \right)$$

(4)

Where, $C_m$ represents the amount of the metals being researched in this study, while $B_n$ represents the background reading for that particular metal. A coefficient of 1.5 is included to accommodate for changes in the background values. In this investigation, the background value for the concentration of metals was determined using the global average concentration given for shale [28].

$I_{geo}$ can be used to classify pollution levels into seven distinct grades or classes [37] (S3 Table).

**Analysis of Toxic Units (TUs).** Relevant toxic units (RTUs) are employed to quantify the potential for significant toxicity caused by heavy metals in soils. Toxic Unit (TU) analysis quantifies the level of environmental hazard by evaluating the concentration of potentially hazardous compounds in soil [38]. When the cumulative toxic units for all soil specimens exceed 4, the toxicity of the hazardous component in the sediment remains at a moderate to high level [39]. The TUs were assessed utilizing the subsequent formula:

$$TU = \frac{C_m}{PEL}$$

(5)

$C_m$ denotes the concentration of heavy metals in soil, while PEL stands for the permissible exposure limit of heavy metals in soil. [40] reported the permissible exposure limit for lead, cadmium, chromium, copper, nickel, and arsenic as 91, 3.5, 90, 197, 36, and 17, respectively. The symbol $\sum$ represents the mathematical operation of summation.

$$\sum TU = TU_{metal_1} + TU_{metal_2} + TU_{metal_3} + \ldots\ldots + TU_{metal_n}$$

(6)

In the case of heavy metals in soil, $\sum$TUs are the result of toxic units [41].

**Ecological risk factor (Erf).** Hakanson [42] established a concept of ecological danger, quantifying the potential ecological consequences of one heavy metal pollutant. This risk can be quantified using the following equation:

$$Erf = T_r \times CF$$

(7)

In this context, $T_r$ represents the noxious response of the heavy metal while CF denotes the degree of pollution. The metals studied were Pb, Cd, Cr, As, Mn, Ni, Cu, Zn, Hg and Co. The hazardous response factors for these metals were computed as 5, 30, 2, 10, 1, 5, 5, 1, 40, and 5 [43], respectively [42] (S3 Table).

**Potential Ecological Risk Index (PERI).** RI is an enhanced iteration of an environmental threat indicator, that functions as an indicator for evaluating the influence of detrimental substances in soils on the ecosystem [44,45]. Here we demonstrate the complete range of hazards presented by the heavy metals found in the soil specimens we analyzed [41,42].

$$RI = \sum_{n=11}^{n} Er$$

(8)

The Potential Ecological Risk Index (PERI) is determined by adding up the various risk factors. It could be categorized into four distinct classes (S3 Table).

## Human health risk assessment

A systematic process used to ascertain the potential health effects of exposure to compounds that cause cancer as well as those that do not is known as human health risk evaluation [46]. The dose-response model was employed to assess the possible health risks to humans in different functional areas. The assessment was employed to calculate the non-cancer dangers to humans through three modes of exposure: ingestion, skin contact, and inhalation. Heavy metals can be introduced into the body by ingestion, inhalation, or dermal absorption when individuals encounter dust settled on surfaces [47,48].

The chronic daily consumption (CDC: mgkg⁻¹day⁻¹) of possibly hazardous heavy metals was calculated using the subsequent formulae [48,49].

$$CDC_{ingestion} = \frac{C_s \times IngR \times EF \times ED}{BW \times AT} \times CF \qquad \text{9(a)}$$

$$CDC_{inhalation} = \frac{C_s \times IngR \times EF \times ED}{PEF \times BW \times AT} \times CF \qquad \text{9(b)}$$

$$CDC_{dermal} = \frac{C_s \times SA \times AF \times ABS \times EF \times ED}{BW \times AT} \times CF \qquad \text{9(c)}$$

The variables in the equation are described comprehensively in a table (S4 Table) [49–51]. If the Hazard Quotient (HQ) or Hazard Index (HI) exceeds one, there is a possibility of non-carcinogenic health impacts. Nevertheless, if the HQ or HI is less than or equal to one, it indicates that there is no proof that exposure to non-carcinogenic metals poses any health risks [48,51,52].

## Analysis of non-carcinogenic risk

The hazard quotient is a measure commonly employed to assess the non-carcinogenic threat. It is calculated by dividing the chronic daily dose of a certain metal by the reference dose [53]. HI represents the cumulative threat of non-carcinogenic heavy metals through three different pathways of exposure for each particular heavy metal [47,53]. However, the subsequent equations are employed to calculate HQ and HI:

$$HQ = \frac{CDC}{Rfd} \qquad (10)$$

$$HI = HQ_{Ingestion} + HQ_{Inhalation} + HQ_{Dermal} \qquad (11)$$

In this research, RfD refers to the reference dose (mgkg⁻¹day⁻¹) for each heavy metal that was examined [54–59]. The reference dose is described comprehensively in a table (S4 Table).

## Assessment of the risk of causing cancer

Regarding the risk of cancer caused by heavy metal exposure, it is possible to evaluate the increased probability of a heavy metal user developing any type of cancer over their entire life [52,60]. Based on the provided computation, the probability of developing cancer can be determined separately for both adults and children [49,53].

$$CR = CDC \times SF \tag{12}$$

$$TCR = \sum CR \tag{13}$$

The slope factor (SF) represents the rate at which a substance causes cancer in a specific population (S4 Table). CR is the amount of a substance that an individual is exposed to, also measured in mg/kg/day. TCR is the cumulative risk of developing cancer from exposure to several substances.

## Results and discussion

This research examined a substantial number of heavy metals. The table presents the descriptive data on the concentration of metals (Table 1).

One-sample Student's t-tests indicated that Cd, Pb, Zn, and Cu concentrations were significantly higher than the shale values ($p < 0.05$), suggesting anthropogenic enrichment. In contrast, Cr, Co, Ni, Mn, and Fe showed significantly lower concentrations than the corresponding shale values ($p < 0.05$). No statistically significant difference was observed for As and Hg ($p > 0.05$).

**Table 1. Heavy metal concentrations in soil samples around the landfill.**

| Sample ID | Metal Concentration (mg/kg) | | | | | | | | | | |
|---|---|---|---|---|---|---|---|---|---|---|---|
| | As | Hg | Cd | Pb | Cr | Zn | Co | Ni | Cu | Mn | Fe |
| 1 | 13.02 | 1.33 | 1.13 | 31.6 | 43.5 | 203.8 | 15.57 | 37.1 | 280.9 | 599.0 | 25852 |
| 2 | 9.86 | 1.21 | 0.528 | 28.8 | 40.9 | 169.3 | 15.57 | 35.8 | 243.1 | 567.2 | 25157 |
| 3 | 7.82 | 0.854 | 0.399 | 21.7 | 37.8 | 204.3 | 13.91 | 29.9 | 189.8 | 424.3 | 21402 |
| 4 | 12.3 | 0.244 | 0.279 | 26.5 | 40.5 | 117.4 | 15.96 | 38.2 | 228.8 | 710.0 | 23506 |
| 5 | 15.1 | 1.51 | 1.76 | 72.1 | 66.2 | 682.4 | 14.16 | 46.4 | 358.4 | 701.2 | 25716 |
| 6 | 13.2 | 0.244 | 0.279 | 54.5 | 44.07 | 155.5 | 17.17 | 36.8 | 233.0 | 492.3 | 27941 |
| 7 | 11.8 | 0.119 | 0.204 | 19.9 | 38.3 | 143.9 | 13.50 | 29.1 | 174.5 | 490.5 | 22200 |
| 8 | 24.8 | 0.609 | 0.274 | 31.4 | 62.9 | 133.7 | 17.26 | 46.8 | 277.0 | 1068.1 | 30931 |
| 9 | 12.07 | 0.607 | 1.27 | 44.6 | 45.2 | 170.1 | 17.77 | 71.5 | 227.9 | 735.8 | 25012 |
| 10 | 7.05 | 0.244 | 1.148 | 51.3 | 44.3 | 425.6 | 16.24 | 35.9 | 348.1 | 370.4 | 23043 |
| 11 | 12.9 | 0.119 | 0.304 | 27.3 | 48.7 | 151.9 | 19.52 | 43.6 | 271.8 | 836.5 | 30902 |
| 12 | 13.6 | 0.608 | 0.673 | 34.2 | 63.9 | 159.1 | 21.99 | 50.6 | 342.8 | 458.1 | 36484 |
| 13 | 11.6 | 0.364 | 0.314 | 34.7 | 51.9 | 231.2 | 19.53 | 40.1 | 272.1 | 775.4 | 29651 |
| 14 | 6.88 | 0.484 | 0.229 | 35.7 | 33.6 | 202.0 | 11.17 | 26.1 | 205.1 | 438.2 | 23627 |
| 15 | 7.96 | 0.609 | 0.284 | 47.7 | 41.2 | 761.2 | 11.12 | 24.7 | 257.9 | 530.9 | 19882 |
| Max | 24.8 | 1.51 | 1.76 | 72.1 | 66.2 | 761.2 | 21.99 | 71.5 | 358.4 | 1068.1 | 36484 |
| Min | 6.88 | 0.119 | 0.204 | 19.9 | 33.6 | 117.4 | 11.12 | 24.7 | 174.5 | 370.4 | 19882 |
| Mean | 12.02 | 0.611 | 0.606 | 37.5 | 46.9 | 260.7 | 16.03 | 39.5 | 260.7 | 613.2 | 26087 |
| SD | 4.40 | 0.441 | 0.487 | 14.08 | 10.06 | 201.0 | 3.02 | 11.6 | 55.79 | 189.6 | 4396 |
| t | −0.86 | 1.85 | 2.43 | 4.80 | −16.60 | 3.19 | −3.80 | −2.82 | 14.98 | −4.84 | −18.60 |
| CV | 36.6 | 72.1 | 80.3 | 37.5 | 21.4 | 77.1 | 18.8 | 29.5 | 21.3 | 30.9 | 16.8 |
| Skewness | 1.69 | 0.901 | 1.31 | 1.10 | 0.972 | 1.95 | 0.12 | 1.40 | 0.427 | 0.949 | 0.909 |
| Kurtosis | 4.80 | −0.166 | 0.688 | 1.12 | −0.146 | 2.70 | −0.13 | 3.16 | −0.502 | 0.747 | 0.731 |
| Shale value[a] | 13 | 0.4 | 0.3 | 20 | 90 | 95 | 19 | 48 | 45 | 850 | 47200 |

[a] [28] SD= Standard Deviation, CV= Coefficient of Variation.

## Distribution of heavy metals

Heavy metal pollution has emerged as a significant threat for environmental safety. Metals are accumulating in our environments due to diverse anthropogenic activities, resulting in an increasing concentration of heavy metals in the soil.

Of the post-transition metals found in the soil specimens, Pb exhibited the highest concentration; over 93% of the samples had concentrations above the shale value of 20 mg/kg. Additionally, the range of Cd concentration in the soil was 0.204 to 1.76 mg/kg, with approximately 60% of samples exceeding the shale threshold of 0.3 mg/kg.

According to [61], the mean concentration of Pb was 14.22 mg/kg around the Sylhet landfill site which was below that of our mean reading of Pb. The average level of Pb was found between 160.34–188.67 mg/kg around the landfills site in Khulna [19]. Pb was detected in the agricultural soil in Bangladesh's Jessore district at 0.26–5.44 mg/kg [62], that is less than what was reported of this investigation (varies:19.99–72.1 mg/kg; average: 37.5 ± 14.08 mg/kg). The cropland around MSW in Guangzhou, located in South China, has a Pb concentration ranging from 11.87 to 21.78 mg/kg [63]. This result is also below that of the findings given in this study. Additionally, it was shown that the top layers of cultivable soils in close proximity to a previous secondary lead smelter in Khulna, Bangladesh, exhibited a significant average Pb concentration of 231 mg/kg. [23], which indicates a pollution level roughly 10 times higher than what was observed in our study. The elevated Pb concentrations in this study are likely attributable to anthropogenic sources, particularly incomplete combustion of waste at the MSW site, which contributes Pb-bearing particulates to the surrounding.

The average concentration of Cd measured 0.606 ± 0.487 mg/kg within the range of 0.204–1.76 mg/kg. For most samples, the amount of Cd present was slightly below the average value of 0.3 mg/kg seen in soils. However, there were a few samples with greater concentrations of Cd. The southwest side of low agricultural land exhibited the greatest concentration of Cd, measuring 1.76 mg/kg in sample ID-5, which is approximately fivefold higher than the shale value. In addition, elevated Cd levels were detected in sample ID-1, sample ID-09, and sample ID-10. The presence of fertilizer and fly ash from landfill may contribute to the rising amounts of Cd in the soil. The presence of Cd in fertilizer and fly ash leads to significant soil pollution [64]. The school field was leveled by using riverside sediment of the Bhairab River, which was very polluted. It is the primary cause of elevated levels of metals in the school field. The Cd values were similar to those reported for the Khulna landfill site, and the Cd was found in allowable limit in Sylhet [19,61].

Zn exhibited the lowest concentration at sample ID-4, measuring 117.4 mg/kg, while it was most concentrated at sample ID-15, with a measurement of 761.2 mg/kg. The primary mechanism for Zn movement in the soil is through solute flux in the soil mass [65]. Therefore, in regions with subtropical climates characterized by abundant rainfall and minimal evapotranspiration, the movement of the metal is mainly influenced by the flow of water inside the soil [66]. This supports the gathering of Zn in the areas of the low-lying slope.

The investigation found that the concentration of Cu varied between 174.5 mg/kg and 358.4 mg/kg, with an average measurement of 260.7 ± 55.79 mg/kg (Table 1). The metal concentration levels at all sites were above the shale value of 45 mg/kg. The sample with the highest concentration of Cu was identified as Sample ID-5, located in the southwest of the treatment plant. Conversely, the sample with the lowest Cu level was identified as Sample ID-7, situated on the north side of the facility (Fig 2).

Batteries, paints, colored glass, waste tires, plastic, and inks in the paper, that are commonly found in municipal solid waste, all include significant amounts of Pb, Cu, Zn, and Cd [12,67,68].

Overall, while Bangladesh is susceptible to As pollution in the present investigation, the average As content in all soil specimens was lower in comparison to the shale measurement of 13 mg/kg. Sample ID-8 exhibited the greatest concentration of As, measuring 24.89 mg/kg, which is approximately double the shale value. The soil is irrigated by river water, which is polluted by nearby industrial effluent. The release of effluents from industrial facilities within the research region, including glass manufacturing plants, perfume production facilities, pharmaceutical factories, and wood processing plants, may contribute to the contamination of soil with arsenic. Sample IDs-1,5,6, and 12 have concentrations proximal to the shale value, and all are agricultural land. These may come from irrigation water and environmental deposition.

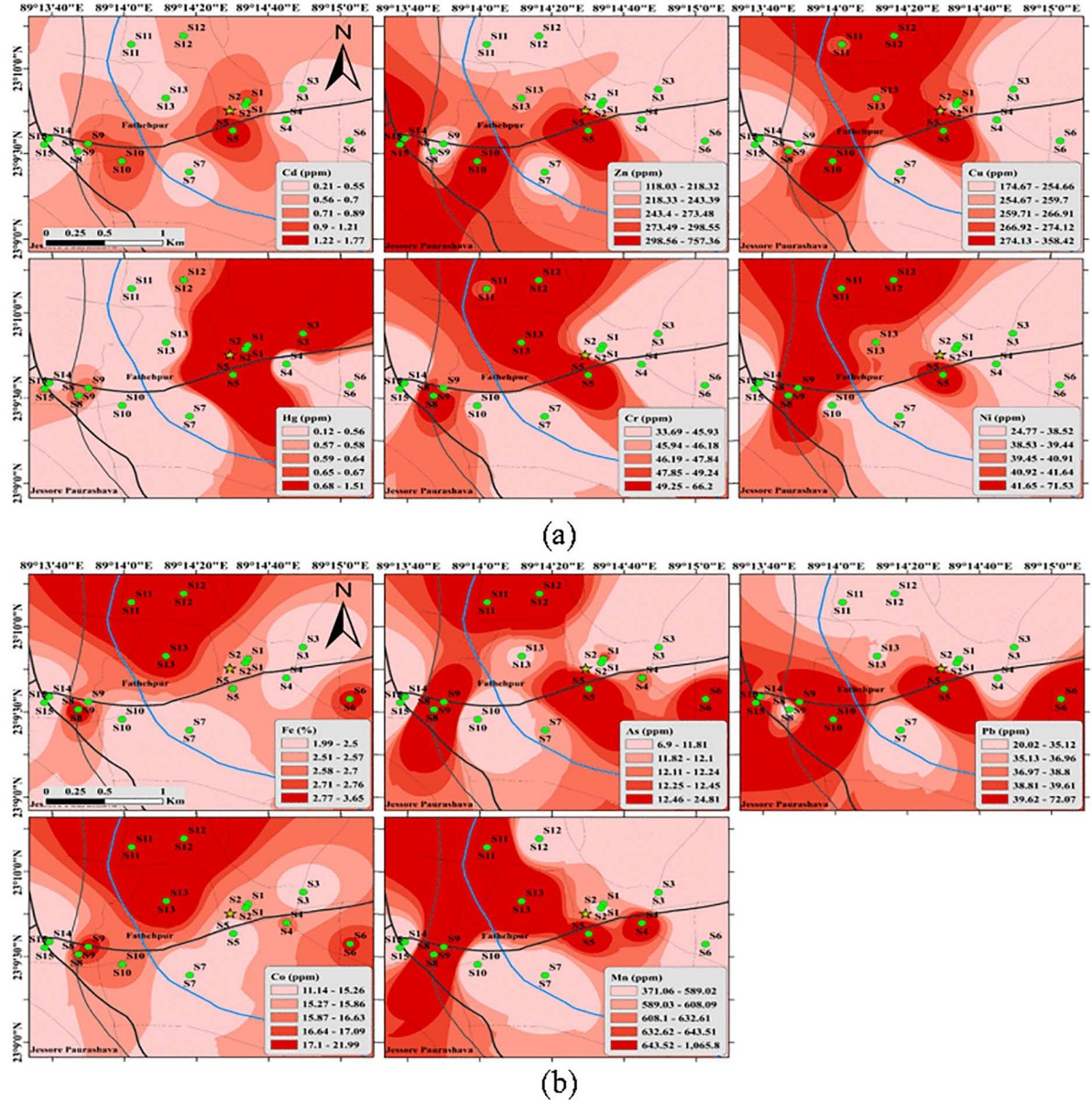

**Fig 2. Spatial distribution of heavy metal (a) Cd, Zn, Cu, Hg, Cr, Ni (b) Fe, As, Pb, Co, Mn concentrations in soil samples around the landfill (Source: Map generated by the authors using ArcMap 10.8 software).**

At Sample ID-10, the quantity of Mn was the lowest, measuring 370.4 mg/kg. Conversely, the highest amount of 1068.1 mg/kg was found in Sample ID-8. The increased concentrations of organic compounds in leachate cause changes in the redox potential of soils impacted by landfill leachate plumes. Therefore, they actively participate in the breakdown of Mn, thereby influencing the movement of this metal [69].

Hg is an extremely hazardous chemical that presents considerable dangers to human beings and other forms of life. Prolonged exposure to any mercury compounds can cause harm to the developing fetus, as well as to the kidneys and

the brain. Moreover, heightened exposure to the harmful effects of mercury can interfere with brain function and lead to symptoms such as tremors, irritation, and alterations in sight and hearing [70].

The Hg values in the region under study vary between 0.119 and 1.51 mg/kg (Table 1). The average concentration is 0.611 ± 0.441 mg/kg, which is higher than the value found in shale. The southwest agricultural field had the highest concentration of Hg at 1.51 mg/kg, which is near the MSW. Conversely, the lowest level of Hg was reported at Sample ID-11, corresponding to a school. The projected concentrations of Hg in municipal solid waste result from the destruction of batteries, fluorescent bulbs, clinical thermometers, and sphygmomanometers.

The average concentrations of Cr, Co, Ni, and Fe were, respectively, 46.9 ± 10.06, 16.03 ± 3.02, 39.5 ± 11.6, and 26087 ± 4396 mg/kg which were lower than the respective values found in shale. However, one sample with the ID-12 showed a higher concentration compared to the shale values. The ID-12 sample is from a paddy field characterized by prolonged wetness throughout the year, potentially leading to the long-term accumulation of contaminants.

This study focused on surface (0–20 cm) soils, which represent the most biologically active layer and primary zone of human exposure. But the potential for vertical migration of heavy metals cannot be overlooked in a tropical monsoon environment. Leachate generated from landfill waste can infiltrate through the soil profile, facilitate the downward transport of metals and may further modify soil physicochemical conditions. Consequently, deeper soil layers and potentially groundwater systems may also be vulnerable to contamination. The present findings primarily reflect surface accumulation patterns, and subsurface investigations are necessary to fully understand the vertical distribution and long-term environmental behavior of heavy metals in this region.

## Comparison of studied soil data with previous studies

The average concentration of heavy metals in the soils of the examined zone has been compared to prior literature reviews conducted in Bangladesh and other prominent landfill areas in the world which is presented in Table 2.

The metal concentration in Sylhet landfill area was below that of the current study [61]. There was a similarity among the metal (Mn, Fe, Cd) concentration with the landfill site in Khulna which may be due to the same source of MSW [19]. When our study is compared to the previous one conducted in Jashore [62], it can be observed that all metal concentrations are lower than this current investigation [71–73]. The difference in metal concentration is influenced by factors such as variations in sampling sites, seasonal changes, the amount of waste deposited in a given year, and their composition. Nevertheless, the levels of dangerous heavy metals in the other municipal waste areas are higher than present study [74–77].

## Soil pollution assessment for heavy metals

The study assessed the prevalence of natural and human-made origins of various metals in the soil around landfill by utilizing contamination indices outlined in the materials and methods section.

**Enrichment factor.** EFs for heavy metals were examined for every specimen in comparison to the crustal proportions (%) of particular metallic components. The enrichment factors for As, Cd, Co, Cr, Cu, Mn, Hg, Ni, Pb, and Zn were 1.67, 3.65, 1.52, 0.94, 10.48, 1.30, 2.76, 1.49, 3.39, and 4.96, respectively (Fig 3a). The average enrichment factors (EFs) for Cd, Hg, Pb, and Zn were above 2. In contrast, the highest EFs for Cd and Cu were above 10, with Zn approaching 19. This indicates that the heavy metals were notably concentrated in these research locations. The average enrichment factor for As, Co, Cr, Mn, Hg, and Ni were below 2 or close to 1, suggesting a deficiency to minimal enrichment. The largest enrichment factor measurements of metals Hg, Pb, Cd, and Cr were identified in sampling ID 5, which is located very close to the municipal used as agricultural land. The average EFs declined in the subsequent sequence: Cu > Zn > Cd > Pb > Hg > As > Co > Ni > Mn > Cr, reflecting the decreasing levels of contamination in the soil of MSW regions. EF can effectively distinguish between natural and anthropogenic causes in the investigation [78].

**Table 2. Comparing the average heavy metal concentration (mg/kg) of our current data with the literature data of soils from landfill site in Bangladesh and the whole world.**

| Sample ID | As | Hg | Cd | Pb | Cr | Zn | Co | Ni | Cu | Mn | Fe | References |
|---|---|---|---|---|---|---|---|---|---|---|---|---|
| Landfill, Jashore | 12.02 | 0.611 | 0.606 | 37.5 | 46.9 | 260.7 | 16.03 | 39.5 | 260.7 | 613.2 | 26087 | Present Study |
| Landfill, Khulna | NA | NA | 0.95 | 122.24 | 20.78 | NA | NA | NA | NA | 595.27 | 27578 | [19] |
| Landfill, Sylhet | NA | NA | 0.15 | 14.22 | NA | NA | NA | NA | 1.63 | 31.23 | 137.79 | [61] |
| Agricultural Land, Jashore | 8.34 | NA | NA | 0.85 | NA | NA | NA | 24.48 | 20.06 | NA | 18000 | [62] |
| Landfill, China | 20.8 | 1.7 | 3.02 | 109 | 101 | 877 | NA | 34 | 241 | NA | NA | [77] |
| Landfill, Denmark | 14 | 0.6 | 6.9 | 373 | 42 | 1020 | NA | 25.7 | 614 | NA | NA | [71] |
| Landfill, Norway | NA | 1.53 | 5.4 | 447 | 21 | 1094 | NA | NA | NA | NA | NA | [75] |
| Landfill, Tehran | NA | NA | 0.07 | 1.02 | NA | NA | NA | 2.83 | 0.86 | 33.92 | 14.92 | [72] |
| Landfill, Brazil | NA | NA | NA | NA | NA | 21.9 | NA | NA | 1.78 | 545 | 44.9 | [73] |
| Guwahati city, Assam, India | NA | NA | NA | NA | 470-1019 | 190 −744 | NA | NA | NA | 361 −874 | NA | [74] |
| Łubna landfill, Poland | NA | NA | 0.035-0.475 | 3.2-31.7 | 1.7-9.9 | 2.3-32.3 | NA | NA | NA | NA | NA | [71] |
| Shale value | 13 | 0.4 | 0.3 | 20 | 90 | 95 | 19 | 48 | 45 | 850 | 47200 | [28] |

NA= Not Available.

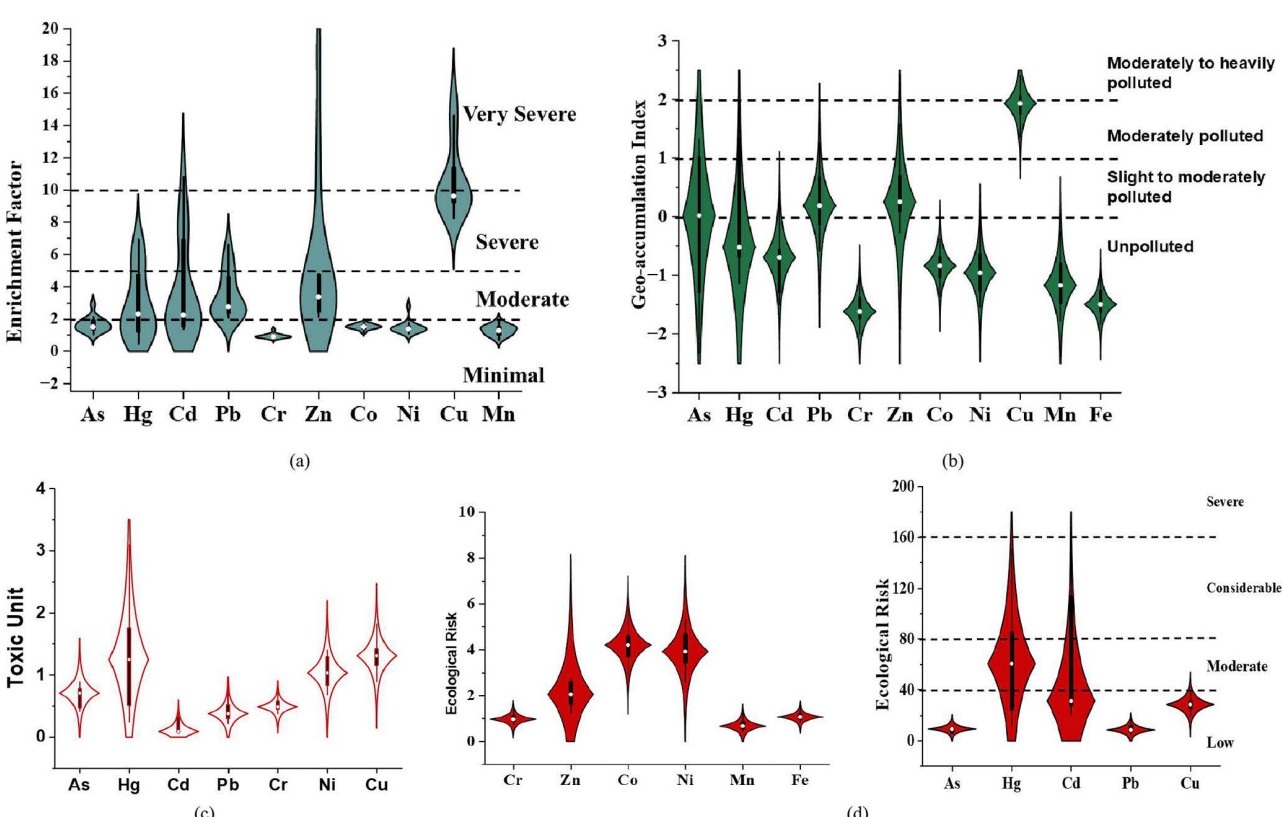

**Fig 3. Boxplot of a) Enrichment Factors (EF) b) Geo-accumulation index ($I_{geo}$) c) Toxic Unit (TU) d) Ecological risk factor (Erf) for metals in the soil of landfill areas.**

**Contamination factor (CF).** S5 Table displays the contamination factors (CFs) and pollutant load index (PLI) data. ID-5 (Agricultural land) had the greatest contamination factor (CF) values for all metals analyzed due to the significant influx of metallic discharge from the MSW. The contamination factor values of Cu exceeded 6 in the soils at most of the sites, indicating a classification of "very high contamination" for this element. The contamination factor values for Cr, Co, Fe, Mn, and Ni in soils at ID-1 indicated low contamination, whereas the contamination factor values for As, Hg, Cd, Pb, and Zn suggested moderate contamination. Cu exhibited the greatest contamination factor values compared to the other ten metals analyzed in the study. Cd and Zn exhibited the second-largest CF values compared to the other ten metals at all sites except ID-8. ID-5 receives municipal wastewater and the agricultural field was filled with the municipal waste ashes, while ID-9 receives river water and filled with river sand. Both sites had high contamination factor values. The Pollution Load Index (PLI) varied between 0.76 and 1.91 as shown in S5 Table. The soils around the MSW were strongly polluted based on the mean PLI value of 1.17. ID-5 exhibited the highest PLI of 1.91 in the research area, suggesting that the sediments at ID-5 were heavily contaminated by the analyzed heavy metals. Sites with PLI values between 1 and 2 should be categorized as moderately contaminated. The PLI sequence was ID-5 > ID-9 > ID-1 > ID-12 > ID-8 > ID-10 > ID-13 > ID-2 > ID-15 > ID-6 > ID-11 > ID-3 > ID-4 > ID-14 > ID-7. CF measurements for Cu and Zn were more than 6 in soils of ID-5, which suggests a "very high contamination" by these metals. The soil samples from ID-5 exhibited considerable contamination for Hg, Cd, and Pb, whereas As, Cr, Co, Fe, Mn, and Ni showed low contamination based on their CF values.

**Geo-accumulation Index ($I_{geo}$).** Fig 3b displays the $I_{geo}$ values of heavy metals found in studied soil. The $I_{geo}$ (geo-accumulation index) of 11 components found in soil specimens ranged from −2–2, respectively. The minimum and maximum values were found for As and Zn respectively. Contamination by metals was discovered in the research region due to the average Geo-accumulation index ($I_{geo}$) being more than one. The elements Cr, Co, Ni, Mn, and Fe with a maximum $I_{geo}$ value of < 0, indicate that they are not contaminated.

The element Zn has an $I_{geo}$ range of −0.27 to 2.41, with an average value of 0.6 (S6 Table). Despite its maximum value exceeding 2, it is classified as slightly to moderately contaminated, however, it can be viewed as a borderline element. Table 2 shows that the shale value for Zn metal in the earth's surface is 95 mg/kg, but the mean value in this study was 260.7 mg/kg.

The soils of ID-5 contained the highest $I_{geo}$ values of the examined metals. The level of Cu and Zn in the soils of ID-5 were very high, indicating moderately to severely polluted in the $I_{geo}$ class of Cu and Zn. The soil samples from ID-5 showed a "moderate pollution" level for the $I_{geo}$ class of As, Cd, and Pb. The $I_{geo}$ values were ranked in the following order: ID-5 > ID-9 > ID-1 > ID-12 > ID-8 > ID-10 > ID-13 > ID-2 > ID-15 > ID-6 > ID-11 > ID-3 > ID-4 > ID-7 (Fig 3b).

**Toxic Unit.** In addition, TU was examined to assess the possible immediate toxicity of heavy metals in soils. TU values and the total of toxic Units (∑TUs) for heavy metals are displayed in Fig 3c. The mean TU values of As, Hg, Cd, Pb, Cr, Ni, and Cu in soils were 0.707, 1.26, 0.178, 0.410, 0.521, 1.09, and 1.32, respectively (S7 Table). Based on the ∑TU findings, the toxicity levels at most of the sampling sites were found to be moderate to high, with ∑TU values exceeding 4. The geographic distribution map of ∑TU indicates that the central region shows high toxicity, whereas the eastern and southern zones of the research region exhibit minimal toxicity (Fig 4).

## Ecological risk assessment

The environmental impact of soil pollution was evaluated by assessing the Er for eleven ecologically hazardous heavy metals. PERI for each location was generated to determine the environmental vulnerability of the environment to these heavy metals. The results of these calculations are displayed in Fig 4 and S8 Table. When compared to other trace metals investigated, Hg poses a moderate ecological danger in the research area, with an average score of 61.17. This is probable because Hg is commonly found in various wastes like batteries, bulbs, etc, which are often dismantled in the area. In addition, Cd also exhibited a low to moderate potential ecological concern. This is because Cd is released during the

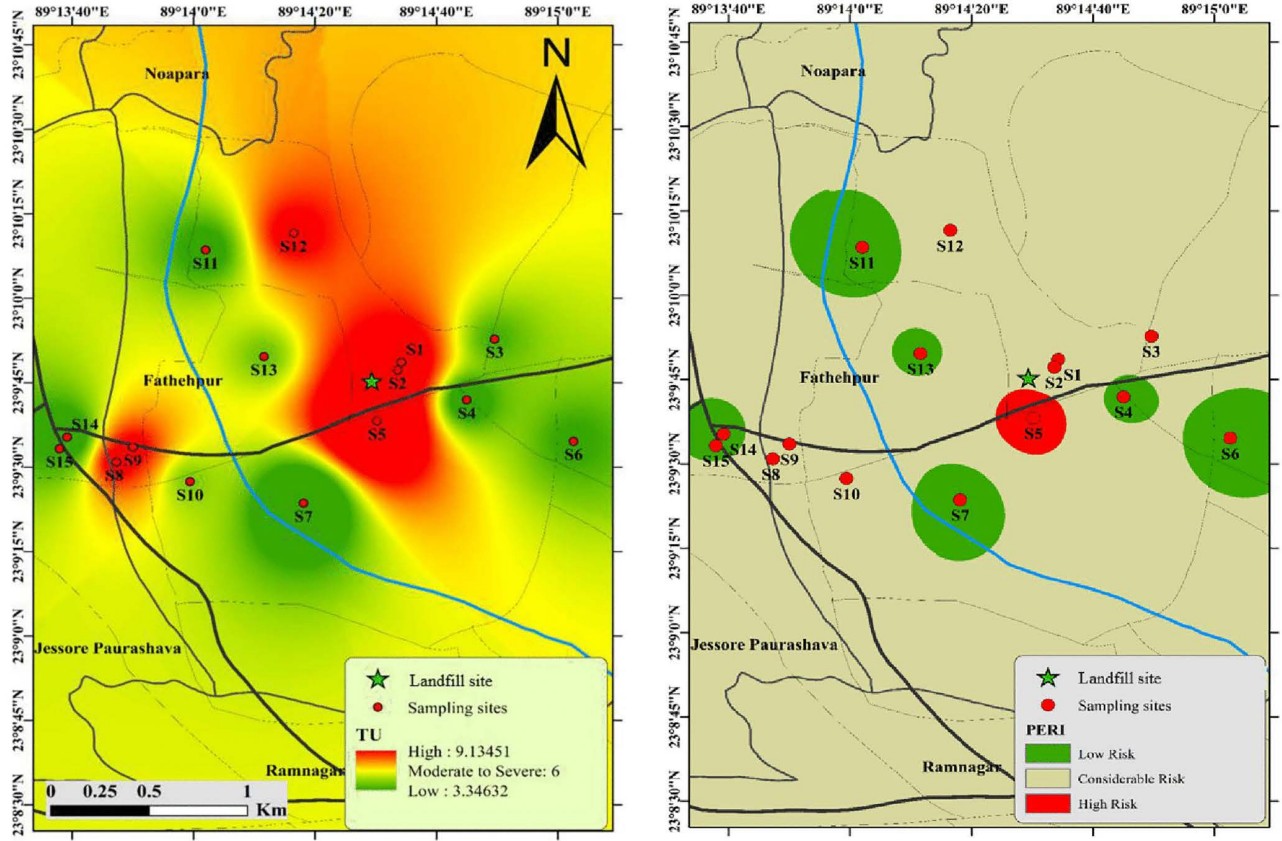

**Fig 4. Spatial distribution maps of ∑TU and Potential ecological risk index (RI) of metals in the soil of landfill areas (Source: Map generated by the authors using ArcMap 10.8 software).**

incineration of MSW such as plastic, Ni-Cd batteries, etc [79]. Nevertheless, the Er values of the remaining heavy metals were determined to be less than 40, as per the classification by the Ref [42]. This indicates that the other trace metals investigated provide either no risk or minimal ecological risks. The mean Er levels for each trace metal are organized as follows: Hg > Cd > Cu > Pb > As > Co > Ni > Zn > Fe > Cr > Mn (Fig 3d).

After examining the geographic distribution, value of the PERI (Fig 4), it becomes evident that the most significant environmental hazards are concentrated in the very close sampling region. The maximum value, 416.44, is observed at point 5, which is situated at a low elevation on the southwest side. Following this, point 1, which is placed at a lower elevation near the abandoned landfills, exhibits a value of 309, which is situated at a low elevation at the east side.

## Health risk

Individuals who work or reside in regions where soils have been contaminated by toxic trace metals may be at risk of exposure to these metals in multiple ways, including direct ingestion of soil particles or consumption of food cultivated in contaminated areas, inhalation, and skin contact. This exposure can lead to a range of health issues. The health risk evaluation approach established by the Ref [53] was employed to investigate the carcinogenic and non-carcinogenic health risks found in soils collected from the Landfill region. The calculated HQ, HI, and TCR values are based on total metal concentrations and assume complete bioavailability; therefore, the actual risks may be lower depending on the bio-accessible fraction of metals.

**Non-carcinogenic risk.** The findings of HQ $_{Ingestion}$, HQ $_{Dermal}$, HQ $_{Inhalation}$, and HI for children and adults, which were obtained from the analysis of heavy metals exposure by various pathways, are displayed in S9A Table in S9 Table. Although the HQ $_{Ingestion}$ values for all metals in adults were below the permissible threshold (< 1), the highest values for Pb ($2.3 \times 10^{-1}$), Cr ($2.82 \times 10^{-1}$), Fe ($6.6 \times 10^{-1}$), Mn ($2.97 \times 10^{-1}$) and Cu ($1.15 \times 10^{-1}$) in children were near the threshold limit. Notably, the HQ $_{ingestion}$ values of As for children (1.06) were above the established safe limit (>1). The HI values of As (1.07) and Zn (4.57) for children are greater than 1, suggesting that the soils in that area pose a significant potential health hazard due to As and Zn contamination. According to [80], an overabundance of Zn has been associated with negative effects such as the loss of brain cells, vomiting, nausea, and a shortage of copper. As exhibits toxic effects on individuals even with short-term exposure. Congenital abnormalities, reproductive complications, and dermatological and vascular illnesses can have enduring effects on the human body, potentially leading to the development of cancer. Arsenicosis, a fatal disease, is caused by the inorganic form of arsenic, a potentially hazardous element, and poses a substantial threat to health [81].

The highest HI values for Pb ($2.35 \times 10^{-1}$), Cr ($3.22 \times 10^{-1}$), Fe ($6.76 \times 10^{-1}$), Cu ($1.16 \times 10^{-1}$), and Co ($1.52 \times 10^{-1}$) in Children were close to the threshold limit. Children are particularly vulnerable than adults to the detrimental impacts of heavy metals obtained through soil. The highest HI values of Zn for adults are close to the threshold limit. The calculated mean HI values of the studied metals in soils, for both adults and Children, were ranked in the following order: Zn>As>Fe > Cr > Mn > Pb > Co > Cu > Ni > Hg > Cd (S9A Table in S9 Table).

**Assessment of carcinogenic risk.** Carcinogenic risk refers to the probability of a person getting cancer throughout their lifetime as a result of being exposed to certain carcinogenic hazards.

The metals with the highest toxicity among the parameters studied, which present a risk of causing cancer, are Pb, Cd, Cr, As, and Ni. Hence, the elevated cancer hazards for both children and adults individually for Pb, Cd, Cr, As, and Ni in the soil for all possible pathways are calculated, as presented in S9B Table in S9 Table. Based on the [49,53], CR and TCR values below $1 \times 10^{-6}$ are regarded insignificant, values between $1 \times 10^{-6}$ and $1 \times 10^{-4}$ are considered acceptable, and values greater than $1 \times 10^{-4}$ are considered possibly hazardous to human.

The maximum CR Ingestion levels for children of Pb ($6.72 \times 10^{-7}$) and Cd ($7.37 \times 10^{-7}$) are within the acceptable range and the value of Cr and As are within the allowed limit. But, the value of Ni ($1.33 \times 10^{-4}$) exceeded the maximum permissible limit. Furthermore, the maximum CR $_{Dermal}$ value of Ni ($1.84 \times 10^{-5}$) is within the allowed limit for children. In addition, the dermal value of other metals as well as all inhalation values for children, were insignificant. The TCR (total cancer risk) value for Ni ($1.52 \times 10^{-4}$) in children exceeded the permissible threshold. The elevated TCR for Ni observed in children can be attributed to site-specific exposure pathways. Notably, school fields in the study area were leveled using contaminated river sediment, which serves as a direct source of exposure. Children are engaged in outdoor activities such as playing and sports in the field. This can lead to increased soil ingestion, inhalation of suspended dust particles or direct dermal contact. However, the values for other metals are within the acceptable range.

However, the maximum ingestion values are seen for As ($2.19 \times 10^{-5}$) and Ni ($7.14 \times 10^{-5}$) while the dermal exposure value was notable only for Ni ($3.34 \times 10^{-5}$). All other ingestion, dermal, and inhalation values for adults were insignificant. The maximum TCR values of Ni ($1.04 \times 10^{-4}$) exceeded the allowable limit for adults, while the level of As ($2.24 \times 10^{-5}$) is within the acceptable range. The values of Pb ($3.59 \times 10^{-7}$), Cr ($3.36 \times 10^{-6}$), and Cd ($4.56 \times 10^{-7}$) are negligible (S9B Table in S9 Table).

The mean TCR values were ranked in descending order as follows: for children, Ni > Cr>As>Cd > Pb; and for adults, As>Cr > Ni > Pb > Cd. As mentioned earlier, children are at greater risk than adults, and the primary ways in which they are affected by the increased risk of cancer are through ingestion and absorption through the skin. The spatial distribution maps indicate that both adults and children in the northern region are at a high risk of developing cancer due to exposure to Cr, Ni, and As. However, in the case of Cd and Pb the southern areas are mostly at a high risk.

## Investigation of the origins of heavy metals pollution

Potential origins of heavy metals were identified using principal component analysis (PCA), cluster analysis (CA), and a Pearson correlation matrix. To investigate the levels of pollution and the origins of heavy metals, three PCs with eigenvalues larger than one were obtained utilizing PCA in this study which is displayed in Table 3 [82].

The initial component (PC1) demonstrated 39.83% of the overall variation with substantial significant loadings on As, Cr, Co, Ni, Cu, Mn, and Fe and had an eigenvalue of 4.38. Situated in the lower riparian zone of the Himalayas, Bangladesh receives substantial sediment loads from rivers originating in the region. These sediments, rich in As and Fe-bearing minerals, are deposited across the floodplains. Since As and Fe can be attributed to rocks in the natural environment, all metals showing significant correlation with these two signify that those metals are mostly of geogenic origin [78]. PC3 showed strongly significant loadings between As and Mn, accounting for 10.52% of the overall variation with an eigenvalue of 1.15, and these elements can originate from natural as well as human-caused sources (Fig 5). Significant positive loadings, nevertheless, were identified in Hg, Cd, Pb, and Cr in PC2, resulting in 28% of the total variation. The grouping is supported by their strong positive correlations ($r > 0.6$; S10 Table), indicating a common source. These statistically significant relationships suggest that these metals are primarily derived from anthropogenic activities, particularly waste burning and landfill-related inputs. The absence of any Hg-bearing rock in the environment of Bangladesh and a strong association with it support the notion that these metals are of anthropogenic origin in the natural environment and the source may be MSW management activities [83].

Additionally, CA was performed utilizing Ward's technique to organize the soil sample sites and heavy metals into distinct categories. The results are displayed as a dendrogram (S1 Fig). The dendrograms revealed the presence of two separate clusters for the metals in soils. The clusters consist of 3 sub-clusters: As-Mn, Cr-Cu, and Co-Fe, which are located within Cluster I (As-Mn-Cr-Cu-Co-Fe-Ni). Co and Fe may share geochemical properties including oxidation states and solubility. Co-occurrence in the environment can help predict metal properties in soils, such as mobility, accessibility to living organisms, and plant absorption. The cluster of Cr and Cu may indicate areas affected by industrial waste generated from metal plating or manufacturing. Ni has a distinct clustering pattern or is included in a bigger cluster at greater heights, suggesting that it is less similar to the other metals or join the clustering process

**Table 3. Principal component analysis of heavy metals in the soils of the landfill area of Bangladesh.**

| Parameters | Coefficient | | |
|---|---|---|---|
| | PC1 | PC2 | PC3 |
| As | 0.730 | −0.317 | 0.506 |
| Hg | 0.241 | 0.583 | 0.372 |
| Cd | 0.474 | 0.712 | −0.078 |
| Pb | 0.352 | 0.774 | −0.098 |
| Cr | 0.927 | 0.115 | 0.057 |
| Zn | −0.016 | 0.869 | 0.101 |
| Co | 0.722 | −0.470 | −0.451 |
| Ni | 0.757 | −0.099 | −0.063 |
| Cu | 0.717 | 0.481 | −0.300 |
| Mn | 0.602 | −0.348 | 0.602 |
| Fe | 0.781 | −0.427 | −0.273 |
| Eigenvalue | 4.38 | 3.08 | 1.15 |
| % of variance | 39.83 | 28 | 10.52 |
| Cumulative | 39.83 | 67.83 | 78.35 |

Every component belongs to rotated component matrix values of trace metals > 0.5

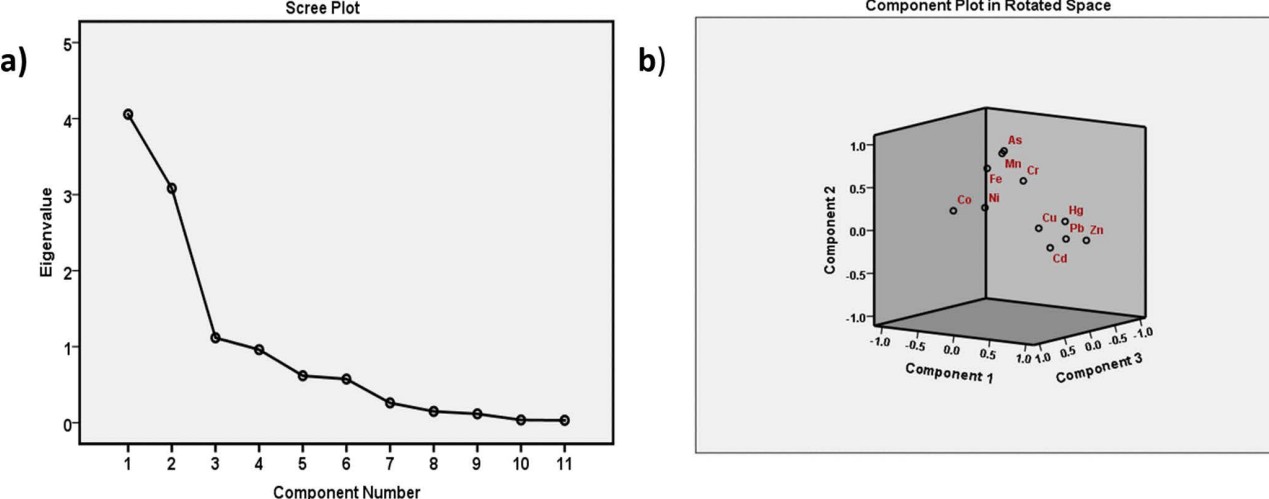

**Fig 5. PCA analysis of heavy metals in the soil of Landfill areas.**

later due to their unique properties. Cluster II comprised two sub-clusters: Hg-Cd and Pb-Zn, which together formed Hg-Cd-Pb-Zn. There was a noticeable disparity in Euclidean distances between the two integrated clusters, suggesting the possibility of distinct origins.

To ascertain the common origin of elements in the MSW area and establish linkages among them, a correlation matrix was computed for trace metals present in the soils. Based on the Pearson correlation coefficients in S10 Table, there was a significant positive association seen among the metals examined.

The correlation between Fe and Co ($r = 0.860$, $p < 0.05$) and between Fe and Cr ($r = 0.860$, $p < 0.05$) were highly significant, suggesting that these metals may be linked to either industrial activity or naturally occurring geological formations where both elements are abundant. The significant link between Cr and Cd ($r = 0.751$, $p < 0.05$) indicates that they likely have common sources, such as metal smelting and the deterioration of copper-based infrastructure, where these elements are frequently found together. The significant link with Pb ($r = 0.695$, $p < 0.05$) indicates shared origins such as discarded batteries, the deterioration of galvanized metal, and the release of pollutants from smelting activities involving both metals.

## Conclusion

A comprehensive analysis was carried out to examine the levels of 11 heavy metals in the soils surrounding the municipal solid waste area in south-western Bangladesh. The average concentration of Hg, Cd, Pb, Zn, and Cu was higher than the corresponding values in shale; On the other hand, the concentration of other metals and metalloids were below the shale values, although they were higher than the shale values at sampling points closer to the landfill. These findings indicate that the soil metal pollution is directly linked to the regular operation of landfill site. While the current contamination level suggests a range from uncontaminated to moderately contaminated, it is clear that if waste combustion in landfill continues at its current rate, contamination levels are likely to grow in the near future. Spatial variability in metal concentrations was observed across sampling sites, influenced by distance from the landfill and topographic relief. This indicates the need for additional investigation, considering the impact of weather conditions, human activity, and expanding the sampling area. Based on multivariate research, it is hypothesized that the burning of waste in landfill is the primary contributor to the presence of trace elements. Furthermore, this information will prove valuable to future policymakers

and environmental researchers who will be studying the ecological areas that have been affected as a result of the open dumping of municipal waste, both locally and globally.

**Limitation**

This study provides a first baseline assessment of heavy metal contamination in soils surrounding the landfill in Jashore. A key limitation of this study is the restriction to surface sampling (0–20 cm). The study focused only on surface soils, which represent the primary exposure layer, particularly under high rainfall conditions. This study relied on global shale values as background references due to the absence of site-specific baseline data and national soil standards in Bangladesh, which may introduce some uncertainty in contamination assessment. Future investigations should incorporate site-specific baselines, depth-profiled sampling and shallow groundwater monitoring to fully characterize vertical metal migration at this site.

## Supporting information

**S1 Table. General characteristics of the sampling locations situated along the landfill.**
(DOCX)

**S2 Table. A. Limits of Detection (LOD) and quantification (LOQ) for elements. B.** Operating Conditions of NexION ICP-MS Instrument and Atomic Absorption Spectrophotometer.
(DOCX)

**S3 Table. Classification of indices for the assessment of soil pollution.**
(DOCX)

**S4 Table. Parameters used to evaluate the human health risk of heavy metals in soil samples.**
(DOCX)

**S5 Table. Metal contamination factors (CFs) and pollution load indices (PLIs) for the soils at all the sites tested along the landfill.**
(DOCX)

**S6 Table. Results of enrichment factors (EF) and geo-accumulation indices ($I_{geo}$) of heavy metals in soils of the landfill area, Bangladesh.**
(DOCX)

**S7 Table. Results of toxic unit (TU) of heavy metals in soils of the landfill area, Bangladesh.**
(DOCX)

**S8 Table. Ecological risk factors (Er) and potential ecological risk indices (RI) of heavy metals in soils of the landfill area, Bangladesh.**
(DOCX)

**S9 Table. A. Assessment of non-carcinogenic human health risks for children and adults in the vicinity of the landfill, Jashore in Bangladesh, considering the potential exposure by ingestion, inhalation, and Dermal pathways. B. Assessment of the carcinogenic risks to human health caused by ingestion, inhalation, and dermal exposure in children and adults residing in the vicinity of the landfill in Bangladesh.**
(DOCX)

**S10 Table. Pearson correlation matrix of heavy metals in the landfill area.**
(DOCX)

**S1 Fig. Hierarchical clustering analysis of metals in the soil of landfill areas.**
(DOCX)

## Acknowledgments

The authors express their gratitude to the authorities of BCSIR for approving this work as a R&D (DLAB-01;2024-2025) and also to the Soil and Environment Research Section, Bangladesh Council of Scientific and Industrial Research (BCSIR), Dhaka, Bangladesh for providing laboratory facilities and logistical support during the research process.

## Author contributions

**Conceptualization:** Md Kamal Hossain, Md. Harunor Rashid Khan.

**Data curation:** Kowshik Das Karmaker.

**Formal analysis:** Sanjida Sultana Santa, Mohammad Moniruzzaman.

**Funding acquisition:** Md Kamal Hossain.

**Methodology:** Sanjida Sultana Santa.

**Resources:** Mohammad Moniruzzaman.

**Software:** Kowshik Das Karmaker.

**Supervision:** Md Kamal Hossain, Md. Harunor Rashid Khan.

**Validation:** Md. Harunor Rashid Khan.

**Writing – original draft:** Sanjida Sultana Santa.

**Writing – review & editing:** Md Kamal Hossain.

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
