## [Decision Letter · Decision Letter 0]

1 Dec 2025

PONE-D-25-27215Heavy metal footprints in landfill-proximate soils of Jashore, Bangladesh: An index-based risk assessmentPLOS ONE

Dear Dr. Hossain,

Thank you for submitting your manuscript to PLOS ONE. After careful consideration, we feel that it has merit but does not fully meet PLOS ONE’s publication criteria as it currently stands. Therefore, we invite you to submit a revised version of the manuscript that addresses the points raised during the review process.

We look forward to receiving your revised manuscript.

Kind regards,

Roshan Babu Ojha

Academic Editor

PLOS ONE

Journal Requirements:

“Ref No-SRG243321 (2024-2025 FY), Ministry of Science and Technology, The Peoples' Republic of Bangladesh”

4. We note that your Data Availability Statement is currently as follows: All relevant data are within the manuscript and in Supporting Information files.

5. We note that Figures 1,2 and 4 in your submission contain map/satellite images which may be copyrighted. All PLOS content is published under the Creative Commons Attribution License (CC BY 4.0), which means that the manuscript, images, and Supporting Information files will be freely available online, and any third party is permitted to access, download, copy, distribute, and use these materials in any way, even commercially, with proper attribution. For these reasons, we cannot publish previously copyrighted maps or satellite images created using proprietary data, such as Google software (Google Maps, Street View, and Earth). For more information, see our copyright guidelines: http://journals.plos.org/plosone/s/licenses-and-copyright.

1. You may seek permission from the original copyright holder of Figures 1,2 and 4 to publish the content specifically under the CC BY 4.0 license.

6. Please ensure that you refer to Figure 5 in your text as, if accepted, production will need this reference to link the reader to the figure.

Additional Editor Comments:

Please provide justification - how 15 samples are sufficient to calculate environmental and human health risk index? How many sub-samples were collected to make 15 composite samples? What is the homogeneity of soil properties within sub-sample collection points? This greatly influences your risk indices.

Reviewers' comments:

Reviewer's Responses to Questions

**Comments to the Author**

1. Is the manuscript technically sound, and do the data support the conclusions?

Reviewer #1: Yes

2. Has the statistical analysis been performed appropriately and rigorously? 

Reviewer #1: N/A

3. Have the authors made all data underlying the findings in their manuscript fully available?

Reviewer #1: No

4. Is the manuscript presented in an intelligible fashion and written in standard English?

Reviewer #1: Yes

5. Review Comments to the Author

Reviewer #1: Heavy metal footprints in landfill-proximate soils of Jashore, Bangladesh: An index-based risk assessment.

Sanjida Sultana Santa 1, Md Kamal Hossain 2,3*, Kowshik Das Karmaker 4, Mohammad Moniruzzaman2,3 , Md. Harunor 3 Rashid Khan 1

Bernardo Sepúlveda.

My observations are:

1. There are several situations in which the plural is used in sentences and should be singular, a couple of examples are: “The average concentrations of Hg, Cd, Pb, Zn, and Cu were higher than, if we are talking about the concept of concentration it should say CONCENTRATION, if we are talking about more than one concentration for each element it would admit the plural; but, it does not seem to be the case; otherwise it should say how many concentrations of each element it is describing. Another example: “While the current contamination levels LEVEL suggest a…”… the “current pollution (I like this word) LEVEL …., etc.

2. In the conclusion it says… “…certain sampling points…”. If these points produce a conclusion, these should be mentioned in some way and in the discussion it should be said what they are or why it is important to be at those points.

3. Regarding the conclusion, it has elements that are not useful for this purpose; I suggest you objectify it to the specific achievements of the work, eliminating phrases that summarize the methodology and discussion. Projections are fine. I propose something like this, it is just an example:

“The average concentration of Hg, Cd, Pb, Zn and Cu was higher than the correspondig values in shale; the concentration of other metals and metalloids was lower than those un shale, although higher than shale level in the …. ¿points?. This results suggest that metal pollution in solils is inked with the regular operation of the landfill. The current level of the contamination ranges from no-polluted to moderately polluted. It is evident that if waste combustión at the landfill continues at the present rate, contamination level are likely to increase in the near future. There was an imbalance in the concentration of metals in the soil samples, varying according to the distance from the landfill and the topography of the terrain. Further research is needed, considering the impact of climatic conditions, human activity and the expansion of the samplig área. It is hypothesised that the burning of waste in landfill is the main factor contributing to the presence of trace elements. This information may prove valuable to researchers and environmental activists studying ecological áreas affected by open dumping of municipal waste, both locally and globally.”

4. In 2.5, there is mention of the software used for the analysis; however, I didn't find the names of the analyses, for example, "Student's t," analysis of variance, Fisher's test, etc. If that's not the case, I haven't mentioned anything; perhaps I did´t see it.

5. I have a reasonable doubt. The results state that certain concentrations are higher or lower than others; however, there are no statistical indicators; they are merely statements based on judgment. I must say this because statistical tests were supposedly applied; therefore, some indicator, such as "p" should have been calculated to confirm the differences between the data.

6. In Table 1... didn't you analysed triplicate samples for each element? I mention this because elsewhere in the text, the value for an element is referred to as "an average." Therefore, if it is an average, it is because there is a statistical error or SD. In these cases the correct nomenclature is xx+xx format. For example, it has been mentioned that 49.6 (unit) is an average of Cr, but the correct value should be 46.9+??.

7. The structure of the tables could be improved, using single-spaced lines and closer columns, specifically in Table 1. Table 3 could be improved, as it repeats the word COEFFICIENT three times; It could be mentioned as a heading on the top line, and the columns could be closer together. All of this would optimize the space occupied.

Coefficient (unit)

A B C

xxxx xxxx xxxx

In general, the work:

a. Addresses an interesting point in current societal issues, such as the impact of garbage dumps.

b. Points to a very important issue such as human health, which is an inalienable right; that is, if this information is used, it would impact the application of people's rights to live in clean environments, etc. I don't need to explain why this is important.

c. Furthermore, it is a local study that could be important to consider for many situations around the world.

d. I believe that the study of anthropogenic pollution is a political duty; we fulfill our duty by making a situation public... I wish the authorities would see this type of information and take action from their positions of power.

e. For me, the work is publishable; but it is also true that the expression must be as good as possible, which is why I recommend that the authors review the work before publishing it.

Atentamente

B. Sepúlveda

Note: Im not used IA for this review.

6. PLOS authors have the option to publish the peer review history of their article (what does this mean?). If published, this will include your full peer review and any attached files.

Reviewer #1: No

---

## [Author Response · Author response to Decision Letter 1]

15 Jan 2026

Response to Reviewers

Manuscript ID: PONE-D-25-27215

Title: Heavy metal footprints in landfill-proximate soils of Jashore, Bangladesh: An index-based risk assessment

Dear Academic Editor and Reviewers,

We sincerely thank you for your valuable comments and suggestions on our manuscript. We have carefully considered each point raised and revised the manuscript accordingly. Below, we provide detailed responses to the reviewers’ comments. The changes made in the manuscript are highlighted with tracked changes in the revised version.

Comments from PLOS Editorial Office:

Authors’ response:

We have carefully reviewed and updated the manuscript to ensure full compliance with all PLOS ONE formatting, style, and file-naming requirements following the official journal templates. The manuscript, figures, tables, references, and supporting files have been revised accordingly.

Authors’ response:

Thank you for this valuable comment. We have added a statement in the Methods section to clarify field access and permit requirements. Soil sampling was conducted in publicly accessible areas surrounding the landfill and on privately owned agricultural land with the voluntary consent of the landowners. No formal permits were required from governmental or institutional authorities, as the study did not involve protected sites or regulated activities. The relevant clarification has now been added to the revised manuscript.

“Ref No-SRG243321 (2024-2025 FY), Ministry of Science and Technology, The Peoples' Republic of Bangladesh”

Authors’ response:

Thanks for the comment. The funders had no role in study design, data collection and analysis, decision to publish, or preparation of the manuscript.

4. We note that your Data Availability Statement is currently as follows: All relevant data are within the manuscript and in Supporting Information files.

Authors’ response:

Thanks for the suggestion. All relevant values are mentioned within the manuscript and supporting information file.

5. We note that Figures 1,2 and 4 in your submission contain map/satellite images which may be copyrighted. All PLOS content is published under the Creative Commons Attribution License (CC BY 4.0), which means that the manuscript, images, and Supporting Information files will be freely available online, and any third party is permitted to access, download, copy, distribute, and use these materials in any way, even commercially, with proper attribution. For these reasons, we cannot publish previously copyrighted maps or satellite images created using proprietary data, such as Google software (Google Maps, Street View, and Earth). For more information, see our copyright guidelines: http://journals.plos.org/plosone/s/licenses-and-copyright.

1. You may seek permission from the original copyright holder of Figures 1,2 and 4 to publish the content specifically under the CC BY 4.0 license.

Authors’ response:

Thank you for the comment. We confirm that Figure 1,2,4 was created by the authors using ArcGIS software (ArcGIS Desktop 10.8, including ArcMap). These datasets are openly licensed and compatible with the CC BY 4.0 license. No copyright or proprietary sources were used.

6. Please ensure that you refer to Figure 5 in your text as, if accepted, production will need this reference to link the reader to the figure.

Authors’ response:

Thanks for the comment. Fig 5 has been cited in the revised manuscript.

Authors’ response:

Thank you for this guidance. In the present review process, the reviewer did not recommend citing any specific previously published works. Therefore, no additional citations were added on this basis. However, we carefully reviewed the manuscript to ensure that all relevant and appropriate literature has already been adequately cited.

Additional Editor Comments:

Please provide justification - how 15 samples are sufficient to calculate environmental and human health risk index? How many sub-samples were collected to make 15 composite samples? What is the homogeneity of soil properties within sub-sample collection points? This greatly influences your risk indices.

Authors’ response:

A total of 15 composite soil samples (3 sub-samples) were collected using a stratified land-use–based sampling approach encompassing agricultural land, school areas, residential zones, and woodland surrounding the municipal treatment plant. These land-use categories represent distinct ecological and human exposure settings relevant to environmental and human health risk assessment. The environmental contamination indices and human health risk indices applied in this study (e.g., CF, Igeo, EF, HQ, HI, CR) rely on average contaminant concentrations rather than inferential statistical testing. Consequently, the reliability of these indices depends primarily on the representativeness of sampling locations and the reduction of small-scale variability, rather than on large sample numbers. Composite sampling is therefore widely used and accepted in soil risk-assessment studies.

Each composite sample was prepared from 3 sub-samples (triplicate) collected at each site from the surface soil layer (0-20 cm). The sub-samples were thoroughly mixed to obtain one homogeneous composite sample, which reduces localized heterogeneity and analytical uncertainty. To ensure homogeneity within each sub-sampling point, samples were collected from areas with uniform land use, similar topography, consistent soil depth, comparable soil texture (clay loam) and color, and no visible recent disturbance. All sub-sampling locations within a site shared similar environmental and management conditions, supporting the assumption of intra-site soil homogeneity.

Reviewers' comments:

Reviewer's Responses to Questions

Comments to the Author

1. Is the manuscript technically sound, and do the data support the conclusions?

Reviewer #1: Yes

2. Has the statistical analysis been performed appropriately and rigorously?

Reviewer #1: N/A

3. Have the authors made all data underlying the findings in their manuscript fully available?

Reviewer #1: No

4. Is the manuscript presented in an intelligible fashion and written in standard English?

Reviewer #1: Yes

5. Review Comments to the Author

Reviewer #1:

Heavy metal footprints in landfill-proximate soils of Jashore, Bangladesh: An index-based risk assessment

Sanjida Sultana Santa 1, Md Kamal Hossain 2,3*, Kowshik Das Karmaker 4, Mohammad Moniruzzaman2,3 , Md. Harunor Rashid Khan 1

Bernardo Sepúlveda.

My observations are:

1. There are several situations in which the plural is used in sentences and should be singular, a couple of examples are: “The average concentrations of Hg, Cd, Pb, Zn, and Cu were higher than, if we are talking about the concept of concentration it should say CONCENTRATION, if we are talking about more than one concentration for each element it would admit the plural; but, it does not seem to be the case; otherwise it should say how many concentrations of each element it is describing. Another example: “While the current contamination levels LEVEL suggest a…”… the “current pollution (I like this word) LEVEL …., etc.

Author’s response:

We thank the reviewer for pointing out the grammatical ambiguity. The text has been revised to clarify singular and plural usage. Specifically, “average concentrations” has been revised to “average concentration of each metal” to indicate one mean value per element, and “contamination levels” have been corrected to “contamination level” where the overall pollution status is discussed.

2. In the conclusion it says… “…certain sampling points…”. If these points produce a conclusion, these should be mentioned in some way and in the discussion it should be said what they are or why it is important to be at those points.

Author’s response:

We thank the reviewer for this helpful comment. In the revised manuscript, the conclusion has been clarified to indicate that elevated concentrations above shale values were observed at sampling locations closer to the landfill. The wording in the conclusion has been revised accordingly to improve clarity and traceability.

3. Regarding the conclusion, it has elements that are not useful for this purpose; I suggest you objectify it to the specific achievements of the work, eliminating phrases that summarize the methodology and discussion. Projections are fine. I propose something like this, it is just an example:

“The average concentration of Hg, Cd, Pb, Zn and Cu was higher than the corresponding values in shale; the concentration of other metals and metalloids was lower than those un shale, although higher than shale level in the …. ¿points?. This results suggest that metal pollution in solils is inked with the regular operation of the landfill. The current level of the contamination ranges from no-polluted to moderately polluted. It is evident that if waste combustión at the landfill continues at the present rate, contamination level are likely to increase in the near future. There was an imbalance in the concentration of metals in the soil samples, varying according to the distance from the landfi

---

## [Decision Letter · Decision Letter 1]

17 Mar 2026

PONE-D-25-27215R1Heavy metal footprints in landfill-proximate soils of Jashore, Bangladesh: An index-based risk assessmentPLOS One

Dear Dr. Hossain,

Thank you for submitting your manuscript to PLOS ONE. After careful consideration, we feel that it has merit but does not fully meet PLOS ONE’s publication criteria as it currently stands. Therefore, we invite you to submit a revised version of the manuscript that addresses the points raised during the review process.

We look forward to receiving your revised manuscript.

Kind regards,

Roshan Babu Ojha

Academic Editor

PLOS One

Journal Requirements:

Additional Editor Comments:

Dear Authors,

Thank you for your patience, we have received major revision from additional reviewer as your original reviewers were not available to track your alterations. Hope to receive your revised version.

Reviewer's Responses to Questions

**Comments to the Author**

1. If the authors have adequately addressed your comments raised in a previous round of review and you feel that this manuscript is now acceptable for publication, you may indicate that here to bypass the “Comments to the Author” section, enter your conflict of interest statement in the “Confidential to Editor” section, and submit your "Accept" recommendation.

Reviewer #2: (No Response)

Reviewer #3: All comments have been addressed

Reviewer #4: (No Response)

2. Is the manuscript technically sound, and do the data support the conclusions?

Reviewer #2: Yes

Reviewer #3: Yes

Reviewer #4: Partly

3. Has the statistical analysis been performed appropriately and rigorously? 

Reviewer #2: No

Reviewer #3: Yes

Reviewer #4: (No Response)

4. Have the authors made all data underlying the findings in their manuscript fully available?

Reviewer #2: No

Reviewer #3: Yes

Reviewer #4: (No Response)

5. Is the manuscript presented in an intelligible fashion and written in standard English?

Reviewer #2: Yes

Reviewer #3: No

Reviewer #4: Yes

6. Review Comments to the Author

Reviewer #2: Please address the following comments in your second revision:

1. While you have justified the use of 15 composite samples based on stratified land use, an open dumping site processing 50 tonnes of waste daily likely creates highly heterogeneous contamination plumes. Please provide a more rigorous technical justification in the "Soil Sampling" section explaining why 15 points are sufficient to capture the full extent of this variability without missing localized "hotspots."

2. Your study is restricted to surface soil (0–20 cm). Given the high organic load and leachate production mentioned in your introduction, please discuss the potential for vertical migration of metals into deeper soil profiles, particularly in the tropical, high-rainfall environment of Jashore.

3. Interpolation Accuracy: In Figures 2 and 4, you present spatial distribution maps for metals and risk indices. You must specify the mathematical interpolation method used (e.g., Kriging, IDW) in Section 2.5 to ensure these maps are not over-extrapolating from a limited dataset.

4. Local Control Sites: Currently, your primary reference for contamination is global average "shale values". It is expected that authors use a local "control" site (soil with similar geological properties but distant from landfill influence). Please explain why a local reference was not established or provide data if available.

5. Please include a table in the Supporting Information listing the Limit of Detection (LOD) and Limit of Quantification (LOQ) for each of the 11 metals analyzed via ICP-MS and AAS. This is essential to verify the precision of your minimum reported values.

6. You report that the Total Cancer Risk (TCR) for Nickel (Ni) in children exceeds permissible thresholds. Given that the school field was leveled using riverside sediment, please expand your discussion on the specific local pathways for this group. Are these children in direct contact with the sediment during daily activities?

7.Your health risk models (HQ, HI, TCR) generally assume 100% bioavailability. Please include a brief discussion on how the bioavailable fraction of these metals might differ from the total concentrations measured, as this is a key factor in screening-level evaluations.

8. In Section 3.6, you attribute metals in PC2 (Hg, Cd, Pb, Cr) to anthropogenic origins such as waste management. Please explicitly reference the specific correlation coefficients from S10 Table in the text to reinforce the statistical link between these elements and waste burning.

I look forward to receiving your point-by-point response and the revised manuscript.

Reviewer #3: After having a close look I understand that the manuscript has been properly addressed following the comments. No other comments from myside.

Reviewer #4: Heavy metals that are defined as toxic to the site, As, Hg, Cd, Pb, Zn, Cu and Mn needs to be compared with either standard background values of the area or with other available standard values. According to the FAO/WHO standard value, only Cu value seems to be in the toxic area, exceeding the normal limit. Therefore, I couldnot find any other convincing fact to expalin the available values of above mentiond elements as in danger zone.

7. PLOS authors have the option to publish the peer review history of their article (what does this mean?). If published, this will include your full peer review and any attached files.

Reviewer #2: **Yes:**Gandhiv Kafle

Reviewer #3: **Yes:**Peiman Zandi

Reviewer #4: No

---

## [Author Response · Author response to Decision Letter 2]

2 Apr 2026

Response to Reviewers

Manuscript ID: PONE-D-25-27215R2

Title: Heavy metal footprints in landfill-proximate soils of Jashore, Bangladesh: An index-based risk assessment

Additional Editor Comments:

Dear Authors,

Thank you for your patience, we have received major revision from additional reviewer as your original reviewers were not available to track your alterations. Hope to receive your revised version.

Response to editor: Thank you for your valuable time and for arranging the additional review. We have carefully revised the manuscript in accordance with the reviewer’s comments and have included a detailed response. We have highlighted the changes in red color which we made based on their feedback in the marked manuscript.

Reviewer's Responses to Questions

Comments to the Author

1. If the authors have adequately addressed your comments raised in a previous round of review and you feel that this manuscript is now acceptable for publication, you may indicate that here to bypass the “Comments to the Author” section, enter your conflict of interest statement in the “Confidential to Editor” section, and submit your "Accept" recommendation.

Reviewer #2: (No Response)

Reviewer #3: All comments have been addressed

Reviewer #4: (No Response)

2. Is the manuscript technically sound, and do the data support the conclusions?

Reviewer #2: Yes

Reviewer #3: Yes

Reviewer #4: Partly

3. Has the statistical analysis been performed appropriately and rigorously?

Reviewer #2: No

Reviewer #3: Yes

Reviewer #4: (No Response)

4. Have the authors made all data underlying the findings in their manuscript fully available?

Reviewer #2: No

Reviewer #3: Yes

Reviewer #4: (No Response)

5. Is the manuscript presented in an intelligible fashion and written in standard English?

Reviewer #2: Yes

Reviewer #3: No

Reviewer #4: Yes

6. Review Comments to the Author

Point-by-point response to reviewer

Reviewer #2: Please address the following comments in your second revision:

1. While you have justified the use of 15 composite samples based on stratified land use, an open dumping site processing 50 tonnes of waste daily likely creates highly heterogeneous contamination plumes. Please provide a more rigorous technical justification in the "Soil Sampling" section explaining why 15 points are sufficient to capture the full extent of this variability without missing localized "hotspots."

Response to reviewer: Thank you for this important and technically insightful comment. We agree that landfill environments can generate highly heterogeneous contamination patterns due to variable waste composition and leachate dispersion. To address this concern, we have strengthened the justification of our sampling design in the revised manuscript (Soil Sampling section). Specifically, a stratified sampling approach was adopted based on land-use categories and directional gradients surrounding the landfill, ensuring coverage of key exposure zones (agricultural, residential, school, and wild land). Each of the 15 sampling points represents a composite of three sub-samples, which reduces microscale variability and improves representativeness. The adequacy of the sampling framework is supported by the moderate to high coefficients of variation (e.g., Cd: 80.3%, Zn: 77.1%), indicating that spatial heterogeneity was effectively captured. Importantly, consistent identification of high-contamination zones (e.g., Site ID-5 and ID-9) across multiple independent indices (CF, PLI, RI) suggests that major contamination hotspots were successfully detected.

2. Your study is restricted to surface soil (0–20 cm). Given the high organic load and leachate production mentioned in your introduction, please discuss the potential for vertical migration of metals into deeper soil profiles, particularly in the tropical, high-rainfall environment of Jashore.

Response to reviewer: Thank you for this valuable comment. We agree that vertical migration of heavy metals is an important consideration, particularly in landfill-affected environments. We have incorporated a detailed discussion in the “Distribution of heavy metals” to address the potential for downward migration of metals. Specifically, we explain that in a tropical monsoon setting such as the study area, characterized by high annual rainfall, landfill-generated leachate can infiltrate the soil profile and facilitate the vertical transport of metals. Additionally, we have acknowledged this as a limitation of the current study (which focuses on surface soils, 0–20 cm) and emphasized the need for future investigations incorporating subsurface soil profiling to better understand vertical distribution and long-term transport mechanisms.

3. Interpolation Accuracy: In Figures 2 and 4, you present spatial distribution maps for metals and risk indices. You must specify the mathematical interpolation method used (e.g., Kriging, IDW) in Section 2.5 to ensure these maps are not over-extrapolating from a limited dataset.

Response to reviewer: Thank you for this suggestion. We have now explicitly stated in Statistical analysis section that Inverse Distance Weighting (IDW) interpolation was applied in ArcMap 10.8 to generate spatial distribution maps (Figures 2 and 4). The choice of IDW was based on its suitability for relatively small datasets and its ability to preserve local variation without excessive smoothing.

4. Local Control Sites: Currently, your primary reference for contamination is global average "shale values". It is expected that authors use a local "control" site (soil with similar geological properties but distant from landfill influence). Please explain why a local reference was not established or provide data if available.

Response to reviewer: We appreciate this important point. Due to logistical constraints and the widespread influence of anthropogenic activities in the surrounding region, identifying a truly unaffected local control site with comparable soil properties was challenging. The use of global shale values [28] enables cross-study comparability and is widely employed in the Bangladeshi literature [19, 61]. The study area and its surroundings are influenced by multiple sources, including landfill activities, agriculture, and nearby industrial operations, making it challenging to identify an uncontaminated reference location. While the use of shale values is a widely accepted approach in similar studies, it may introduce some uncertainty. Therefore, future research should incorporate site-specific background values from geologically similar but minimally impacted areas to improve accuracy.

5. Please include a table in the Supporting Information listing the Limit of Detection (LOD) and Limit of Quantification (LOQ) for each of the 11 metals analyzed via ICP-MS and AAS. This is essential to verify the precision of your minimum reported values.

Response to reviewer: Thanks for the comment. The Limit of Detection (LOD) and Limit of Quantification (LOQ) level for each of the 11 metals analyzed has been added in the supplementary S2a table.

6. You report that the Total Cancer Risk (TCR) for Nickel (Ni) in children exceeds permissible thresholds. Given that the school field was leveled using riverside sediment, please expand your discussion on the specific local pathways for this group. Are these children in direct contact with the sediment during daily activities?

Response to reviewer: Thank you for this valuable comment. We have expanded the discussion in carcinogenic risk to clarify site-specific exposure pathways. The school field was leveled using contaminated river sediment, which is frequently used by children for daily outdoor activities such as playing and sports. Thus, direct ingestion of soil/dust, dermal contact, and incidental inhalation of resuspended particles are likely dominant exposure pathways for children. These behaviors significantly increase exposure frequency and explain the elevated TCR for Ni observed in children compared to adults.

7.Your health risk models (HQ, HI, TCR) generally assume 100% bioavailability. Please include a brief discussion on how the bioavailable fraction of these metals might differ from the total concentrations measured, as this is a key factor in screening-level evaluations.

Response to reviewer: We thank the reviewer for this insightful comment. We have added a statement in the Health Risk Assessment section acknowledging that HQ, HI, and TCR calculations assume 100% bioavailability, and that actual risks may be lower depending on the bio-accessible fraction of metals.

8. In Section 3.6, you attribute metals in PC2 (Hg, Cd, Pb, Cr) to anthropogenic origins such as waste management. Please explicitly reference the specific correlation coefficients from S10 Table in the text to reinforce the statistical link between these elements and waste burning.

Response to reviewer: Thanks for the comment. This correlation has been added in the PCA analysis section.

I look forward to receiving your point-by-point response and the revised manuscript.

Reviewer #3: After having a close look I understand that the manuscript has been properly addressed following the comments. No other comments from myside.

Response to reviewer: Thank you very much for your careful evaluation of the manuscript and for acknowledging that the revisions have adequately addressed the previous comments. We sincerely appreciate your time and constructive feedback, which has helped improve the quality of our work.

Reviewer #4: Heavy metals that are defined as toxic to the site, As, Hg, Cd, Pb, Zn, Cu and Mn needs to be compared with either standard background values of the area or with other available standard values. According to the FAO/WHO standard value, only Cu value seems to be in the toxic area, exceeding the normal limit. Therefore, I could not find any other convincing fact to explain the available values of above-mentioned elements as in danger zone.

Response to reviewer: Thank you for your valuable and constructive comment. We fully agree that the comparison of heavy metal concentrations with appropriate background or standard values is essential for accurately interpreting contamination levels. However, in the present study, establishing site-specific background values was challenging due to the widespread influence of anthropogenic activities (e.g., landfill operations, agriculture, and nearby industrial sources) across the study area and its surroundings. These factors limited our ability to identify a truly uncontaminated reference site with comparable geochemical characteristics. Furthermore, there is currently no established national soil quality standard for Bangladesh, and available FAO/WHO guideline values do not comprehensively cover all the analyzed metals in this study. Therefore, we adopted global shale values as background references, which is a widely accepted approach in similar environmental assessments and has been applied in previous studies conducted in Bangladesh (Refs. [19, 61]). We acknowledge that the use of generalized shale values may introduce some level of uncertainty. In light of this, a statement has been included in the revised manuscript highlighting this limitation. Future studies are recommended to incorporate site-specific baseline values from geologically similar but minimally impacted areas to enhance the accuracy and robustness of contamination assessment.

7. PLOS authors have the option to publish the peer review history of their article (what does this mean?). If published, this will include your full peer review and any attached files.

Do you want your identity to be public for this peer review? For information about this choice, including consent withdrawal, please see our Privacy Policy.

Reviewer #2: Yes: Gandhiv Kafle

Reviewer #3: Yes: Peiman Zandi

Reviewer #4: No

---

## [Decision Letter · Decision Letter 2]

6 May 2026

Heavy metal footprints in landfill-proximate soils of Jashore, Bangladesh: An index-based risk assessment

PONE-D-25-27215R2

Dear Dr. Hossain,

We’re pleased to inform you that your manuscript has been judged scientifically suitable for publication and will be formally accepted for publication once it meets all outstanding technical requirements.

Kind regards,

Roshan Babu Ojha

Academic Editor

PLOS One

Additional Editor Comments (optional):

Congratulations!!! and thank you for your hard work.

Reviewers' comments:

Reviewer's Responses to Questions

**Comments to the Author**

1. If the authors have adequately addressed your comments raised in a previous round of review and you feel that this manuscript is now acceptable for publication, you may indicate that here to bypass the “Comments to the Author” section, enter your conflict of interest statement in the “Confidential to Editor” section, and submit your "Accept" recommendation.

Reviewer #2: All comments have been addressed

2. Is the manuscript technically sound, and do the data support the conclusions?

Reviewer #2: Yes

3. Has the statistical analysis been performed appropriately and rigorously? 

Reviewer #2: Yes

4. Have the authors made all data underlying the findings in their manuscript fully available?

Reviewer #2: No

5. Is the manuscript presented in an intelligible fashion and written in standard English?

Reviewer #2: Yes

6. Review Comments to the Author

Reviewer #2: Please make data available publicly, with a public weblink. Follow journal guidelines for this. It is one of the major criteria of the journal.

7. PLOS authors have the option to publish the peer review history of their article (what does this mean?). If published, this will include your full peer review and any attached files.

Reviewer #2: **Yes:**Gandhiv Kafle

---

## [Editor Report · Acceptance letter]

PONE-D-25-27215R2

PLOS One

Dear Dr. Hossain,

I'm pleased to inform you that your manuscript has been deemed suitable for publication in PLOS One. Congratulations! Your manuscript is now being handed over to our production team.

Kind regards,

on behalf of

Dr. Roshan Babu Ojha

Academic Editor

PLOS One